# Exceptionally High Heat Flux Needed to Sustain the Northeast Greenland Ice Stream

Silje Smith-Johnsen[1], Basile de Fleurian[1], Nicole Schlegel[2], Helene Seroussi[2], and Kerim Nisancioglu[1,3]

[1]Department of Earth Science, University of Bergen, Bjerknes Centre for Climate Research, Norway
[2]Jet Propulsion Laboratory, California Institute of Technology, Pasadena, California, USA
[3]Centre for Earth Evolution and Dynamics, University of Oslo, Oslo, Norway

**Correspondence:** Silje Smith-Johnsen (silje.johnsen@uib.no)

**Abstract.** The Northeast Greenland Ice Stream (NEGIS) currently drains more than $10\%$ of the Greenland Ice Sheet area, and has recently undergone significant dynamic changes. It is therefore critical to accurately represent this feature when assessing the future contribution of Greenland to sea level rise. At present, NEGIS is reproduced in ice sheet models by inferring basal conditions using observed surface velocities. This approach helps estimate conditions at the base of the ice sheet, but cannot
be used to estimate the evolution of basal drag in time, so it is not a good representation of the evolution of the ice sheet in future climate warming scenarios. NEGIS is suggested to be initiated by a geothermal heat flux anomaly close to the ice divide, left behind by the movement of Greenland over the Icelandic plume. However, the heat flux underneath the ice sheet is largely unknown, except for a few direct measurements from deep ice core drill sites. Using the Ice Sheet System Model (ISSM), with ice dynamics coupled to a subglacial hydrology model, we investigate the possibility of initiating NEGIS by inserting heat
flux anomalies with various locations and intensities. In our model experiment, a minimum heat flux value of $970\,\mathrm{mW\,m^{-2}}$ located close to The East Greenland Ice-core Project (EGRIP) is required locally to reproduce the observed NEGIS velocities, giving basal melt rates consistent with previous estimates. The value cannot be attributed to geothermal heat flux alone and we suggest hydrothermal circulation as a potential explanation for the high local heat flux. By including high heat flux and the effect of water on sliding, we successfully reproduce the main characteristics of NEGIS in an ice sheet model without using
data assimilation.

## 1   Introduction

The Greenland Ice Sheet (GrIS) displays large spatial variations in surface velocity, with a few fast-flowing outlets draining most of the interior (Rignot and Mouginot, 2012). It is therefore critical to capture the complex flow pattern of GrIS in models used for future sea level projections. Recent developments in ice sheet models such as efficient parallel computation (Khroulev
and PISM-Authors, 2015), better representation of flow equations (Larour et al., 2012), detailed basal topography (Morlighem et al., 2014) and the inclusion of subglacial hydrology have contributed to greatly improve the representation of this spatially varying flow (Aschwanden et al., 2016). In addition to these advances, inversion for basal friction using surface velocities has proved to be a powerful tool (Morlighem et al., 2013), and models are now able to capture most of the complex flow pattern of the ice sheet. Inversions are useful to capture present day velocity, but mask information that is needed to evolve these

conditions in time. Therefore, we cannot fully rely on inversions for future projections, as basal conditions may evolve as a result of a changing climate and in turn influence ice dynamics.

The Northeast Greenland Ice Stream (NEGIS) drains more than 10 % of GrIS and is exceptional by displaying high velocities all the way to the ice divide (Rignot and Mouginot, 2012). Despite its large impact on the GrIS mass balance, NEGIS is not accurately represented in ice sheet models without inverting for basal friction (Goelzer et al., 2018). Aschwanden et al. (2016)

simulated NEGIS in the Parallel Ice Sheet Model, capturing high velocities using a simple hydrology model, however, lacking the far inland onset of the ice stream. Beyer et al. (2018) used the basal melt rates from the model by Aschwanden et al. (2016) in a more sophisticated hydrology model to reproduce NEGIS in the Ice Sheet System Model (ISSM). They capture the high velocity flow of the outlets well, but the representation of the transition areas outside of the main trunk are more diffuse compared to the observed values. These studies illustrate how we are getting closer to reproducing present day NEGIS in ice

sheet models. However, the characteristic clearly defined shear margins and high velocities upstream at the onset of the ice stream are still lacking.

To understand why high upstream velocities are not reproduced in models, one must look into how the ice stream is initiated. The origin of NEGIS has been explained by a geothermal heat flux (GHF) anomaly left behind by the passage of the Icelandic plume (Fahnestock et al., 2001; Rogozhina et al., 2016; Martos et al., 2018; Alley et al., 2019). Interpretation of radar

data points to unusually high basal melt rates at the head of the ice stream, corresponding to an exceptionally high GHF of $970 \, \mathrm{mW \, m^{-2}}$ (Fahnestock et al., 2001; Macgregor et al., 2016; Alley et al., 2019; Keisling et al., 2014). A local increase in GHF intensifies basal water production and potentially enhances basal sliding. Unfortunately, GHF maps for Greenland display a large spread of values (Rogozhina et al., 2012; Shapiro and Ritzwoller, 2004; Fox Maule et al., 2009; Martos et al., 2018; Rogozhina et al., 2016; Greve, 2019). These large uncertainties in the estimates of the GHF have been shown to dominate

the uncertainty on the ice flux in this region (Smith-Johnsen et al., 2019). In addition, the GHF maps are coarse and may not capture local anomalies like the one suggested to exist at the head of NEGIS (Fahnestock et al., 2001; Macgregor et al., 2016; Alley et al., 2019). Accurately capturing such a feature and explicitly representing the effect of high melt rates on basal sliding, is key to reproduce the distinct velocity pattern of NEGIS in ice sheet models.

Here, we study the impact of the presence and intensity of a mantle plume, at the head of NEGIS on the ice flow structure.

We do not suggest the presence of a mantle plume, but rather use an existing mantle plume model to generate feasible GHF scenarios in the model sensitivity study. We use a sophisticated hydrology model (de Fleurian et al., 2014, 2016) coupled to ice dynamics in the Ice Sheet System Model (ISSM; Larour et al., 2012) to capture the influence of enhanced basal melt on ice dynamics. We first describe the models and different plume experiments. Finally, we present and discuss resulting basal conditions and surface velocities corresponding to the various plume configurations.

**Table 1.** Definitions and values of variables in the subglacial hydrology model

| Description | Unit | Value |
|---|---|---|
| effective pressure | Pa | |
| compressibility of water | $\text{Pa}^{-1}$ | $5.04 \times 10^{-10}$ |
| leakage factor | m | $1 \times 10^{-9}$ |
| inefficient compressibility | $\text{Pa}^{-1}$ | $1 \times 10^{-8}$ |
| inefficient porosity | | 0.4 |
| inefficient thickness | m | 20 |
| inefficient transmitivity | $\text{m}^2\,\text{s}^{-1}$ | 0.002 |
| efficient compressibility | $\text{Pa}^{-1}$ | $1 \times 10^{-8}$ |
| efficient porosity | | 0.4 |
| efficient initial thickness | m | 0.005 |
| efficient collapsing thickness | m | $8 \times 10^{-5}$ |
| efficient maximal thickness | m | 5 |
| efficient conductivity | $\text{m}^2\,\text{s}^{-1}$ | 25 |

## 2 Methods

### 2.1 Ice Flow Model

To simulate the NEGIS ice flow, we apply the model configuration from Schlegel et al. (2013, 2015) further developed and coupled to a subglacial hydrology model by Smith-Johnsen et al. (2019). We use the Ice Sheet System Model (Larour et al., 2012), a 3D thermomechanical ice flow model, and explicitly represent the effect of high melt rates on subglacial hydrology (de Fleurian et al., 2014, 2016), which provides the effective pressure ($N$, the difference between ice overburden pressure and water pressure at the bed) that controls basal sliding through a linear friction law (Cuffey and Paterson, 2010):

$$\tau_b = -\alpha^2 N \mathbf{v}_b, \tag{1}$$

where $\tau_b$ is the basal drag, $\alpha$ basal friction coefficient and $\mathbf{v}_b$ the basal velocity. The hydrology model takes the basal melt rates as input, and computes the effective pressure. Nodes with no basal melt are given an effective pressure equal to the ice overburden pressure. The hydrology model consists of two porous sediment layers, representing the inefficient and efficient drainage system. The efficient drainage system is activated when $N$ reaches zero, and may be deactivated as the water is evacuated and $N$ increases again. Definitions and values of variables in the subglacial hydrology model are given in Table 1. The hydrology model and its implementation in ISSM are described in detail in de Fleurian et al. (2014, 2016).

For the thermal model we rely on the enthalpy formulation by Aschwanden et al. (2012), implemented in ISSM (Seroussi et al., 2013) with surface temperatures from Ettema et al. (2009) and GHF from Fox Maule et al. (2009). In addition we use a mantle plume module in ISSM to create elevated GHF anomalies (Seroussi et al., 2017). Ice is treated as a purely viscous

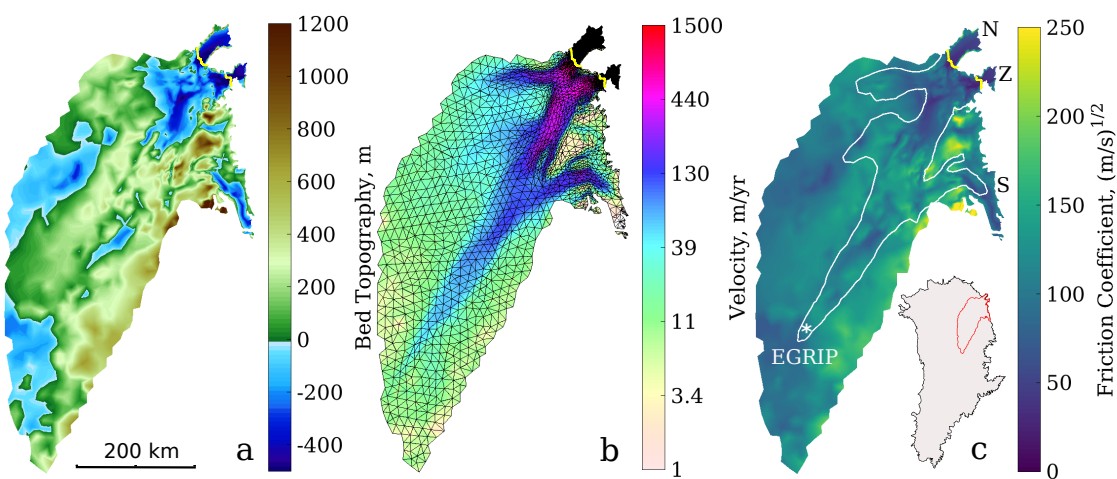

**Figure 1.** (a) bed topography from BedMachine (Morlighem et al., 2014) interpolated onto the model mesh, (b) InSAR-derived surface velocities (Rignot and Mouginot, 2012) and anisotropic model mesh refined in areas with high velocity gradients, (c) friction coefficient as a linear function of bed topography (Eq. 3) used in Eq. 1. The white contour shows the area of the NEGIS with observed surface velocity of $50\,\mathrm{m\,yr^{-1}}$ and the star shows the position of the East Greenland Ice-Core Project (EGRIP). N, Z and S indicate the outlets of the ice stream; 79N, Zachariæ and Storstrømmen respectively. The yellow line in all panels represent the grounding line and the inset map in the lower right corner shows Greenland with the model domain outlined in red.

incompressible material (Cuffey and Paterson, 2010), with viscosity, $\mu$, defined as:

$$\mu = \frac{B}{2\dot{\epsilon}_e^{\frac{n-1}{n}}}, \tag{2}$$

where $B$ is the temperature dependent ice hardness varying with depth, $n$ is Glen's flow law exponent and $\dot{\epsilon}_e$ is the effective
strain rate.

Basal topography is from BedMachine (Morlighem et al., 2014) (Figure 1a) and we apply submarine melt rates under the floating ice (Rignot et al., 2001). For the stress balance equation, we use a 3D Higher-Order approximation (Pattyn, 2003). Our model domain consists of 9974 horizontal elements, ranging from 1 km in areas with high velocity gradients to a maximum of 15 km at the ice divide (Figure 1b). We use linear P1 elements to solve the stress balance equations and quadratic P2 elements
for the thermal analysis, in order to capture sharp temperature gradients, despite using only five layers (Cuzzone et al., 2018).

We aim to represent the observed NEGIS velocity pattern in an ice sheet model without inverting for the basal friction coefficient. However, to initialize the hydrology model, we do simulate the present day ice stream by inferring basal friction from present-day velocities (Figure 1b). The basal melt rates from this simulation are used to initialize the subglacial hydrology model, which we run for $150$ years in order to reach an equilibrium in terms of water pressure. The resulting effective pressure
field computed by the hydrology model, $N$, is used in the friction law (Eq.1), and kept constant in time. Finally, we run a $4\,\mathrm{kyr}$ simulation with the basal condition generated by the hydrology model to provide steady state surface velocities. Note that we

**Table 2.** Mantle Plume parameter overview for the plume experiments

| Parameter | Description | Value | Unit |
|---|---|---|---|
| mantleconductivity | mantle heat conductivity | 2.5 | $\mathrm{W\,m^{-3}}$ |
| nusselt | nusselt number, ratio of mantle to plume | 500000 | |
| dtbg | background temperature gradient | 0.013 | $\mathrm{degree\,m^{-1}}$ |
| plumeradius | radius of the mantle plume | varying | m |
| topplumedepth | depth of the mantle plume top below the crust | 5000 | m |
| bottomplumedepth | depth of the mantle plume base below the crust | varying | km |
| crustthickness | thickness of the crust | 1 | m |
| uppercrustthickness | thickness of the upper crust | 1 | m |
| uppercrustheat | volumic heat of the upper crust | $1.33 \times 10^{-6}$ | $\mathrm{W\,m^{-3}}$ |
| lowercrustheat | volumic heat of the lower crust | $2.7 \times 10^{-7}$ | $\mathrm{W\,m^{-3}}$ |

do not use the friction coefficient, $\alpha$, from the inversion in the forward ice flow simulation, as it is only used to initialize the subglacial hydrology model.

Previous modelling studies lack sharp velocity gradients defining NEGIS (Aschwanden et al., 2016; Beyer et al., 2018). To capture this we let the basal friction coefficient, $\alpha$, depend linearly on the bed elevation using the following equation:

$$\alpha = \min(\max(1, 0.13 \times bed + 100), 250), \tag{3}$$

where $100\,(\mathrm{m\,s^{-1}})^{1/2}$ is the mean value of the inversion alpha used in (Smith-Johnsen et al., 2019), and we cap the values between 1 and $250\,(\mathrm{m\,s^{-1}})^{1/2}$. The factor 0.13 is tuned to approximately match the observed velocities at the grounding line of 79N. The resulting friction coefficient, $\alpha$, is shown in Figure 1c. We argue that low lying topography will have more marine sediments, and thus a softer and less resistive bed, allowing high velocities of the outlet glaciers. A similar approach with basal shear stress defined as a function of bed elevation was previously used by Åkesson et al. (2018) and by Aschwanden et al. (2016). Our simple friction relationship is supported by observations, as bed topography roughness for the NEGIS region shows a pattern inversely correlated with bed elevation (Cooper et al., 2019). This relation might however not hold on smaller scales under the NEGIS trunk where the till distribution is independant of the bed geometry (Christianson et al., 2014). Some alternatives to using this parameterisation are given in the discussion section of this paper.

## 2.2 Experiments

In order to capture the high upstream velocity of NEGIS, we alter the GHF by simulating a mantle plume close to the head of the ice stream, at the onset of fast flow (Seroussi et al., 2017). The mantle plume module in ISSM computes the GHF, given the plume parameters in Table 2. To disentangle the effect of the mantle plume we run a Ctrl simulation without a mantle plume,

**Table 3.** Overview of mantle plume parameters, modelled GHF and friction parameters.

| Simulation | Position | Radius (km) | Depth (km) | max GHF (mW m$^{-2}$) | $\alpha$ ((m s$^{-1}$)$^{1/2}$) | N (MPa) |
|---|---|---|---|---|---|---|
| Ctrl | no plume | no plume | no plume | no plume | varying | modelled |
| plume970 | center | 50 | 5000 | 970 | varying | modelled |
| plume677 | center | 50 | 3000 | 677 | varying | modelled |
| plume836 | center | 50 | 4000 | 836 | varying | modelled |
| plume909 | center | 50 | 4500 | 909 | varying | modelled |
| plume970SW | SW | 50 | 5000 | 970 | varying | modelled |
| plume970SE | SE | 50 | 5000 | 970 | varying | modelled |
| plume970NE | NE | 50 | 5000 | 970 | varying | modelled |
| plume970NW | NW | 50 | 5000 | 970 | varying | modelled |
| plume494 | center | 300 | 3000 | 494 | varying | modelled |
| plume594 | center | 200 | 2500 | 594 | varying | modelled |
| plume775 | center | 100 | 2000 | 775 | varying | modelled |
| plume792 | center | 200 | 3000 | 792 | varying | modelled |
| noHydro | no plume | no plume | no plume | no plume | varying | approximated |
| Ctrl-uni | no plume | no plume | no plume | no plume | 90 | modelled |
| plume970-uni | center | 50 | 5000 | 970 | 90 | modelled |

using only the GHF from Fox Maule et al. (2009). This GHF map is ranging from $40\,\mathrm{mW\,m^{-2}}$ in the north-west to $77\,\mathrm{mW\,m^{-2}}$ in the north-east below the Storstrømmen outlet, with an average value of $54\,\mathrm{mW\,m^{-2}}$.

In our main experiment, plume970, the plume parameters were chosen to generate a GHF anomaly coherent with the magnitude of the GHF anomaly hypothesized by Fahnestock et al. (2001). The resulting GHF anomaly is $\sim 50\,\mathrm{km}$ in diameter with
a maximum GHF value of $970\,\mathrm{mW\,m^{-2}}$ (Table 3), and we position it directly underneath the EGRIP deep ice core drilling site (Figure 1c).

To determine the minimum GHF needed to initiate the onset of NEGIS close to the ice divide, we compute three alternative plume configurations with lower intensity. We obtain the lower GHF by decreasing the bottom plume depth parameter to 4500, 4000 and 3000 km for simulation plume909, plume836 and plume677, respectively (Table 3). Additionally, we compute
four plume configurations where we change the position of the plume. We move the plume970 75 km to the south-west, south-east, north-east and north-west in the plume970SW, plume970SE, plume970NE, plume970NW experiments, respectively (Table 3). To investigate the influence of the area of the mantle plume, we compute four plume configurations with larger area, compensated for by a smaller heat flux. To obtain this we increase the plume radius to values of 100–300 km, and decrease the bottom plume depth to values of 2000–3000 km, resulting in the experiments plume494, plume594, plume775 and plume792
(Table 3).

Finally, to investigate the influence of our friction coefficient distribution, we run three additional simulations. First, we run a simulation without modelled effective pressure, but instead using effective pressure approximated to hydrostatic pressure,

commonly used in ISSM (no Hydro, Table 3). Then we run two simulations with a uniform friction of $\alpha = 90 \, (\text{m s}^{-1})^{1/2}$; one without a plume (Ctrl-uni, Table 3) and one with the $970 \, \text{mW m}^{-2}$ plume (plume970-uni, Table 3).

## 3  Results

In the Ctrl simulation we use the GHF from Fox Maule et al. (2009) (Figure 2a), and the corresponding basal melt rates are shown in Figure 2f. Melt rates at the head of the ice stream (at EGRIP) are $1$–$2 \, \text{mm yr}^{-1}$, and the highest basal melt rates ($600 \, \text{mm yr}^{-1}$) occur at the grounding line of Zachariæ, with surface velocities reaching $1500 \, \text{m yr}^{-1}$. Friction is the dominating heat source in the fast flowing regions, and melt rates thus increase with increasing velocities towards the grounding line. Low melt rates in regions with high velocity are due to low lying bed topography causing low basal drag and hence less frictional heat. The effective pressure for the Ctrl experiment is shown in Figure 2k, and the values increase upstream toward the ice divide as ice thickness increases and basal melt decreases. The lowest values of effective pressure coincide with low bed elevation in the main trunk, 100 km upstream of the grounding line.

The resulting velocity field for the Ctrl simulation captures the main features of NEGIS: the three outlets with high velocities across the grounding lines and sharp shear margins (Figure 2p). The northern branch feeding into 79N is slower and less defined than in the observed velocities, and the velocities of Storstrømmen are also slower than observed. Velocities of the floating tongues of 79N and Zachariæ are not well represented, and floating shelves are not shown here. The western branch, feeding into the main trunk of NEGIS, shows a more diffuse pattern with higher velocities than observed.

To evaluate how well the model simulations reproduce the observed velocity pattern, we plot the $50 \, \text{m yr}^{-1}$ velocity contour (black contour in Figure 2), and compare how far upstream this contour reaches (in kilometres from the ice divide) relative to the observed velocity (white contour in Figure 2). The modelled velocity contour in the Ctrl reaches 305 km from the ice divide (Figure 2p), and thus further downstream than the observed velocity (120 km, Figure 2a,f,k). The Ctrl simulation does not capture the characteristics of NEGIS; with high upstream velocities close to the ice divide.

To capture the upstream velocities, we enhance the GHF locally at the onset of the ice stream in the plume970 simulation, to reach the maximum magnitude proposed by Fahnestock et al. (2001). The addition of the mantle plume results in high GHF, with values up to $970 \, \text{mW m}^{-2}$, rapidly decreasing to the values used in Ctrl (Figure 2b) within a radius of less than 100 km. High geothermal heat leads to high basal melt rates, with $\sim 100 \, \text{mm yr}^{-1}$ above the plume (Figure 2g), compared to $1$–$2 \, \text{mm yr}^{-1}$ in the Ctrl experiment. The increase in basal melt rates causes a reduction in effective pressure to 1.2 MPa directly above the plume, resulting in a local floatation fraction (ratio of water pressure over overburden pressure) of 0.95. The resulting velocity field in the plume970 experiment is similar to the Ctrl experiment, except for the higher velocities simulated at the head of the ice stream. In the plume970 simulation the $50 \, \text{m yr}^{-1}$ velocity contour reaches 131 km from the ice divide (black contour Figure 2q), which is close to the observed 120 km. However, the spatial pattern upstream is more diffuse and the ice stream is wider than observed. The Storstrømmen outlet shows higher velocities relative to the Ctrl, but still lower than observed. The 79N and Zachariæ outlets, on the other hand, display higher velocities than observed. Overall, with this

approach, we capture most of the characteristics of NEGIS, although the ice stream is more diffuse and displays velocities slightly higher than the observations.

To determine whether a lower GHF may induce a similar high velocity pattern, we run three simulations with a less intense mantle plume. Figure 2c-e show the GHF values computed by rising the plume depth to 3000, 4000 and 4500 km, respectively, obtaining maximum basal melt rates of $\sim 70$ (Figure 2j), $\sim 85$ (Figure 2i) and $\sim 95 \, \mathrm{mm \, yr^{-1}}$ (Figure 2h). The modelled effective pressure for the three plumes (Figure 2m-o) result in slower velocities than the plume970, with $50 \, \mathrm{m \, yr^{-1}}$ velocity contours reaching to 253, 245 and 210 km from the ice divide, respectively (Figure 2r-t). This shows that GHF values of 677, 836 and 909 $\mathrm{mW \, m^{-2}}$ produce weaker ice stream signatures than observed, and given our model set-up, are not sufficient to induce the upstream fast flow of NEGIS.

To investigate the sensitivity of the position of the plume in plume970, we moved the plume 75 km to the south-west, south-east, north-east and north-west (Figure 3). The computed GHF distribution is shown in (Figure 3a-d) and the basal melt rates are of the same magnitude as in the plume970. The computed effective pressure for south-west and south-east (plume970SW and plume970SE, Figure 3i, j) have minimum values of 3.2 and 2.9 MPa above the plume, which are not sufficient to initiate fast flow (Figure 3m, n). When the plume is located further downstream, the effective pressure reaches lower values (Figure 3k, l) and the ice stream flows faster than in plume970 (Figure 3o, p). However, with the $50 \, \mathrm{m \, yr^{-1}}$ contour only reaching 204 km from the ice divide. The plume970NE induces the fastest flow, and the plume970NW creates an interesting double branched ice stream starting from the ice divide. The experiments in Figure 3 indicate that the elevated heat required to initiate the NEGIS in our model must be located close to EGRIP.

To determine whether a lower GHF value over a larger area could induce high upstream velocities, we investigate the influence of four weaker plumes with larger plume radii (Figure 4). The weakest, but most extensive plume (plume494, Figure 4a), produces basal melt rates of maximum $51 \, \mathrm{mm \, yr^{-1}}$ (Figure 4e), resulting in a large area of low effective pressure (minimum 0.2 MPa; Figure 4i). The corresponding surface velocity for the plume494 displays a faster and wider ice stream (Figure 4m) relative to the observations. Plume594 gives basal melt rates of $60 \, \mathrm{mm \, yr^{-1}}$ (Figure 4f) and the ice stream becomes wide, reaching all the way to the ice divide (Figure 4n). The plume775 is twice the size of the plume970 (Figure 4c), and with melt rates of $\sim 75 \, \mathrm{mm \, yr^{-1}}$ over a larger area (Figure 4g), the velocity of the ice stream (Figure 4o) is similar to the plume970. However, the $50 \, \mathrm{m \, yr^{-1}}$ velocity contour reaches too close to the ice divide and the ice stream is wider than the observed one. The plume792 produces melt rates of $\sim 75 \, \mathrm{mm \, yr^{-1}}$ (Figure 4d), resulting in velocities similar to the plume594 (Figure 4p). This shows that plumes with a restricted extent, $\sim 50 \times 50 \, \mathrm{km}$, produce model results more consistent with the observed flow behaviour in the upstream reaches of NEGIS.

Finally, we investigate the influence of varying the parameters in the friction law (Eq. 1), presented in Figure 5. The noHydro simulation with an effective pressure approximated to the hydrostatic pressure shows very little resemblance to the observed NEGIS (Figure 5a), with too slow velocities. The simulation with a uniform friction coefficient and no mantle plume, captures the main feature of NEGIS (Ctrl-uni, Figure 5b); with a main trunk, the northern branch and three outlets, with fastest flow in Zachariæ. However, the velocity pattern is more diffuse than the observed (Figure 5e). The high upstream velocities are better captured in the simulation with plume970 and a uniform friction (plume970-uni, Figure 5c). For plume970-uni, high

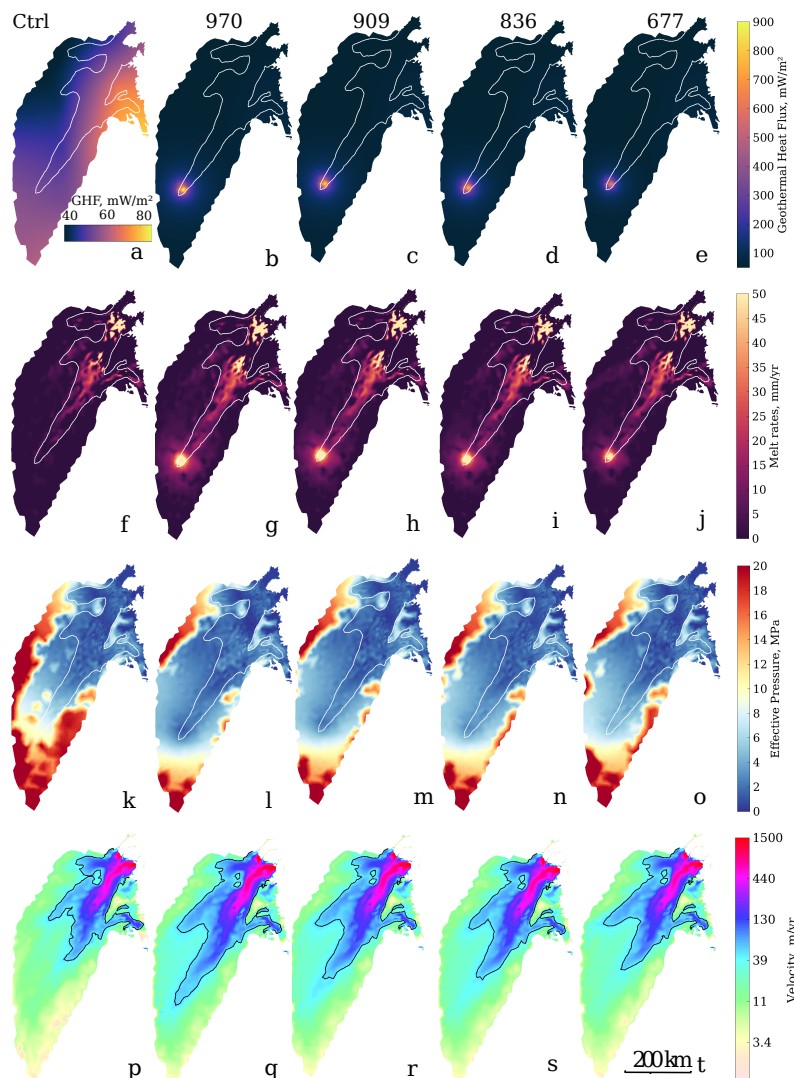

**Figure 2.** Model results for the Ctrl and the plume677, plume836, plume909 and plume970 simulations. a-e show the modelled GHF (note the different color scale for Ctrl) and f-j shows the corresponding basal melt rates, forcing the hydrology model which computes the corresponding effective pressure (k-o) and finally the resulting surface velocity (p-t). White lines show the $50\,\mathrm{m\,yr^{-1}}$ observed velocity contour, and black lines show the $50\,\mathrm{m\,yr^{-1}}$ modelled velocity contour.

velocities reach slightly closer to the ice divide than the plume970, but the velocities of the main trunk are less confined than in experiment plume970 (Figure 5d) and the observations (Figure 5f).

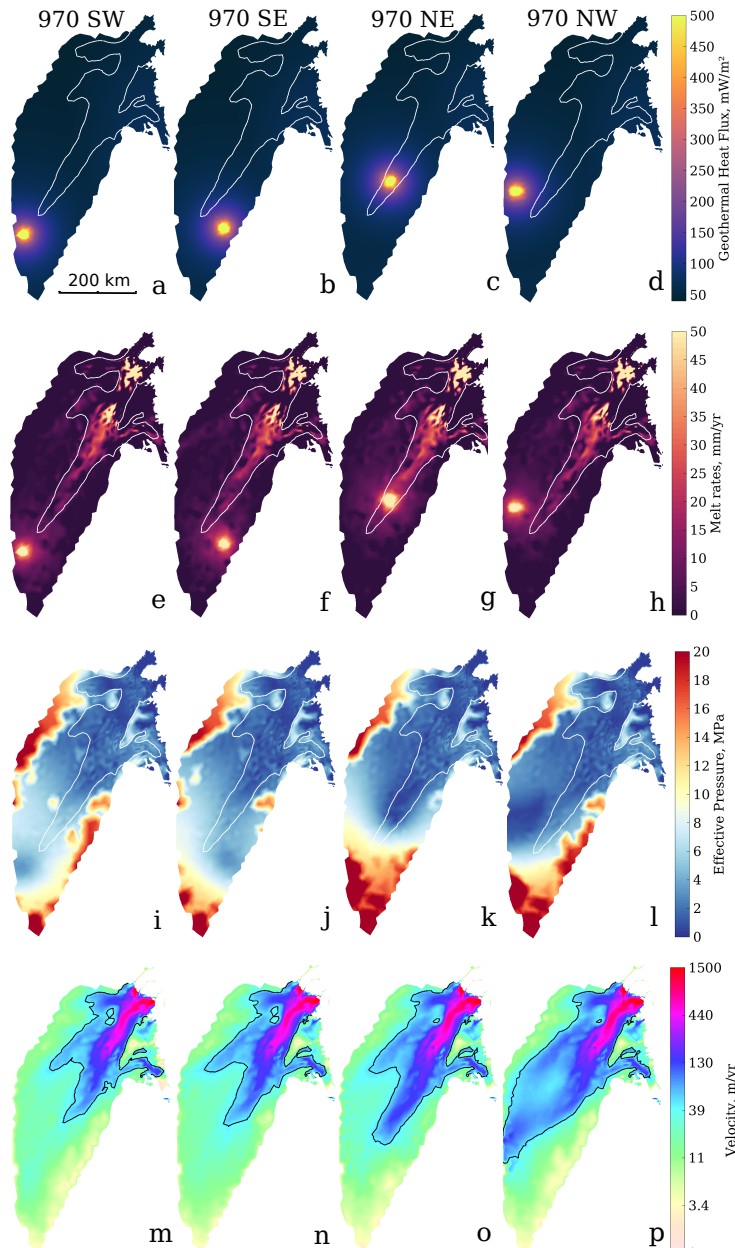

**Figure 3.** Model results from the sensitivity simulations investigating the position of the mantle plume by moving the plume970 75 km. First column shows results from plume970SW, with a plume 75 km to the south-west, second column represents 970 SE Plume, third represents plume970NE and the last column is plume970NW. a-d show the GHF, e-h the resulting basal melt rates, i-l the computed effective pressure and m-p the modelled surface velocity. White lines show the $50\,\mathrm{m\,yr^{-1}}$ observed velocity contour, and black lines show the $50\,\mathrm{m\,yr^{-1}}$ modelled velocity contour.

## 4    Discussion

Most of the spatial velocity pattern of NEGIS is represented in our Ctrl run, apart from the upstream one third of the main trunk. This indicates that the downstream area of the NEGIS catchment is largely controlled by topography, while the upstream area is controlled by its basal conditions, which is in agreement with Keisling et al. (2014). The Ctrl simulation captures the main outlets and the observed "snake" shaped velocity pattern of the trunk. High velocities coincide with low lying bed elevation. However, we do not capture the high velocity of Storstrømmen, or the floating tongues of Zachariæ and 79North outlets. This could be caused by the simple friction coefficient approach not being representative of these areas, where basal properties display a more complex pattern.

We performed experiments with various mantle plume configurations introduced at the head of NEGIS, to assess if the presence of an anomalously high GHF can explain the pattern of ice flow of this region. The different plume configurations vary in intensity, position and extent. In the Ctrl simulation we use present day surface velocity and GHF from Fox Maule et al. (2009). Without the presence of a plume, the GHF does not reach more than $54\,\mathrm{mW\,m^{-2}}$ and leads to underestimating velocities in the upstream part of the catchment. These low values of GHF are not sufficient to initiate the onset of NEGIS close to the ice divide. By testing with four mantle plume configurations of increasing intensity (Figure 2), we find that the GHF (GHF) needed to induce the observed upstream velocity of NEGIS in our model, is $\sim 970\,\mathrm{mW\,m^{-2}}$.

A GHF of $970\,\mathrm{mW\,m^{-2}}$ is consistent with the maximum value presented in Fahnestock et al. (2001); Keisling et al. (2014) for regions in proximity of EGRIP, where the plume970 is located. It also compares well to the anomaly modelled by Macgregor et al. (2016) in the trunk of NEGIS but do not include the high GHF that they find upstream. These GHF values are imposed based on basal melt estimates from radar internal stratigraphy. Our modelled basal melt rates ($\sim 100\,\mathrm{mm\,yr^{-1}}$) are thus consistent with their proposed values. By directly comparing the basal melt rates of our plume970 experiment to the basal melt rate estimates from Macgregor et al. (2016) in Figure 6, it can be seen that our plume produces a basal melt pattern that matches the position, extent and values of the north-eastern branch of their anomaly. The sensitivity simulations in Figure 3m,n show that more than $970\,\mathrm{mW\,m^{-2}}$ is needed to initiate high velocity, when the plume is located further upstream in a region with thicker ice relative to downstream. This suggest that the area of high basal melt estimated by Macgregor et al. (2016) in the trunk of NEGIS is probably more consequential than the larger melt anomaly that they modeled closer to the divide.

The GHF at the head of NEGIS is suggested to be high due to lithospheric thinning as a results of the Iceland plume passage (Rogozhina et al., 2016; Martos et al., 2018). However, $970\,\mathrm{mW\,m^{-2}}$ is an extremely high GHF value, ten to twenty times higher than the values suggested by GHF models for Greenland (Shapiro and Ritzwoller, 2004; Fox Maule et al., 2009; Martos et al., 2018; Rogozhina et al., 2016). Greve (2019) derived GHF values for five deep ice core bore holes in Greenland, using the SICOPOLIS model (SImulation COde for POLythermal Ice Sheets; www.sicopolis.net), such that the simulated and observed basal temperatures match. This resulted in a local elevated GHF anomaly around NGRIP of $135\,\mathrm{mW\,m^{-2}}$, located at the ice divide $\sim 150\,\mathrm{km}$ away from the head of NEGIS. Our GHF anomaly has a magnitude seven times higher than Greve (2019) and three times as high as the highest current GHF observations in Greenland (Rysgaard et al., 2018). In summary, the plume970 produces a basal melt pattern with magnitude and extent in line with previous estimates from the radar data for the region

within the $50\,\text{m\,a}^{-1}$ isoline, however there is a large discrepancy between the necessary GHF to produce this melt and the GHF estimates for Greenland.

To explain the high GHF value of $970\,\text{mW\,m}^{-2}$ we need to investigate processes that may locally elevate the GHF. Alley et al. (2019) and Stevens et al. (2016) explained high GHF in this region by the passing of the Iceland plume, leaving behind partly molten rock that may have migrated up in response to glacial-interglacial cycles, as the crust is loaded and unloaded. A study showed that glacial rebound may have caused young intraplate volcanism in Greenland, despite the old age of the tectonic plate and no mantle plume present (Uenzelmann-Neben et al., 2012). The plume passage could have lead to shallow magma emplacements, that may feed hydrothermal systems, causing hot fluid percolation that enhances high heat transport to the base of the ice sheet (Stevens et al., 2016; Alley et al., 2019; Mordret, 2018). It is important to note that the term GHF is defined as the heat flux from the Earth's interior as a pure conductive heat transfer. Hence, the $970\,\text{mW\,m}^{-2}$ heat flux can not be explained by GHF alone, but rather surface heat flow from locally elevated GHF due to advective heat transfer from the processes mentioned above (Artemieva, 2019).

Comparing the velocity field in the plume970 experiment to previous studies without inversion shows that combining a basal hydrology model with an elevated GHF at the head of NEGIS captures the observed high, confined, upstream velocities of the NEGIS. The simulations in Goelzer et al. (2018) show that the ice flow models capturing the upstream onset of NEGIS all rely on inversions to initialize the basal drag in the simulations (Elmer/Ice, ISSM, BISICLES, GRISLI and f.Etish). The models without inversion, underestimate the velocities in the upper part of NEGIS catchment and lack the sharp velocity gradients. Aschwanden et al. (2016) simulated the high upstream velocity of NEGIS without inverting for basal conditions in PISM, but their simulation lacks the clearly defined main trunk and underestimates the high upstream velocity. Beyer et al. (2018) further improved the simulation by using a subglacial hydrology model to compute effective pressure, which allowed higher velocities in the outlets. However, high upstream velocities are still lacking, similar to our Ctrl simulation. The two latter studies used GHF from Shapiro and Ritzwoller (2004), which proposed slightly lower values at the head of NEGIS compared to the values of Fox Maule et al. (2009) used in our study.

Beyer et al. (2018) used the same friction law as we use in ISSM, but with a uniform friction coefficient. We tested a uniform friction coefficient, which lead to a more diffuse ice stream (Figure 5b,c), but with more confined outlets compared to the Beyer et al. (2018) study. The difference can be explained by different basal melt rates used as input, and different hydrology models. In order to capture sharp gradients in the velocity field, we find it important that the areas without any basal melt have effective pressure equal to the ice overburden pressure.

We invert for basal friction to get the basal melt rates that are used to initialize the subglacial hydrology model, and the model is then free to evolve. We do not use the inverted friction in the forward ice flow simulation, instead we use the simple friction coefficient from Eq. 3. To investigate whether the modelled velocity pattern is caused by the effective pressure distribution or the friction coefficient, we run the simulation 'no Hydro', where the effective pressure is approximated to the hydrostatic pressure, commonly used in ISSM. The modelled velocity pattern (Figure 5a) does not resemble the observed, and we conclude that including the subglacial hydrology model is responsible for the improved velocity pattern in Ctrl and plume970. By using

our friction coefficient distribution, combined with initializing with present-day basal melt from velocity observations, both the Ctrl and plume970 experiments display velocity patterns similar to the observations (Figure 5d, e).

     The middle western branch of the ice stream displays too high velocity in both the Ctrl and plume970 experiments, correlating with low lying bed elevation (Figure 1). Too high velocities in this region were also modelled by Aschwanden et al. (2016) using PISM and a similar bed elevation dependent friction law. When performing additional simulations with the GHF

values from Martos et al. (2018) this branch becomes more pronounced in velocity (not shown here). This may indicate that the GHF values in this region of Greenland are even lower than Martos et al. (2018) and Fox Maule et al. (2009), and the glacier base is frozen to the ground. This region is recognized as "uncertain" in the synthesis of Greenland's basal thermal regime by Macgregor et al. (2016). Other explanations for too high velocities in this branch may be a higher bed roughness, errors in the bed topography or "sticky spots".

Given the model configuration, an exceptionally high heat flux of $970\,\mathrm{mW\,m^{-2}}$ is needed to reproduce NEGIS. We acknowledge that this value may be overestimated due to uncertainties and assumptions in our model set-up, and we discuss these in the following sections. We use a simple friction law linearly dependent on effective pressure, and are aware that the results are likely to change with a different choice of friction law. For example, in the friction law used in the MISMIP+ experiments (Asay-Davis et al., 2016; Tsai et al., 2015), effective pressure is included only where the coulomb criterion is met, normally

a few km upstream of the grounding line. This may result in a smaller dynamic response from the mantle plume in the slow upstream regions of NEGIS. However, the use of a non linear friction law may enhance the sensitivity of the ice dynamics to effective pressure, also upstream, as we compute low effective pressure above the plume. This implies that the use of a non-linear friction law may result in a lower GHF needed to sustain NEGIS in a model.

     By using a coarse model mesh we may underestimate the softening occurring due to strain heating in the shear margins,

and hence overestimate the lateral drag. Refining the mesh and inducing damage softening of the ice in the shear margins (Bondzio et al., 2017), would decrease the lateral drag. In this case, the observed high upstream velocity of NEGIS may have been reproduced with higher basal drag and hence lower GHF. The underestimation of modelled ice softness may also explain why our modelled upstream velocity field is wider and more diffuse than the observed.

     In the simulations where we investigate the influence of an increased plume radii (Figure 4), we show that lower values of

GHF can induce even faster flow, when the plume is more extensive (Figure 4). However, with a larger mantle plume the ice stream becomes wider, and does not match the observed velocity of NEGIS (Figure 5e). The basal melt pattern of Macgregor et al. (2016) in Figure 6 consists of two melt anomalies near EGRIP. It would be interesting to investigate the velocity response of two weaker elevated GHF anomalies closely located. There is also room for improvement of the model in the treatment of the shear margin or the use of a non linear friction law (Gagliardini et al., 2007; Schoof, 2005). Both those improvements

would lead to sharper transition from slow to fast velocities and might allow a plume with a larger radius.

     We parametrize the friction coefficient with a simplified estimate linearly dependent on the bed elevation. In other studies this coefficient is inverted for by matching observed surface velocity, producing low values in the main trunk of NEGIS (Smith-Johnsen et al., 2019). By lowering the friction in the main trunk, we may reproduce fast flow with a lower GHF value. However, this would make the friction coefficient relate to the velocity, which we are trying to avoid. The bed topography used is from

BedMachine (Morlighem et al., 2014), so datasets used to create this map impact the choice of friction. A uniform lowering of the friction coefficient, also outside the trunk, would increase velocities all over the domain, hence we would loose the sharp velocity gradients and overestimate the outlet velocity even further. Additionally, the modelled ice surface in the Ctrl experiment is lower than the observed (Scambos and Haran, 2002), and a uniform reduction of friction will enhance this mismatch. We do not observe a local depression in the surface topography above the $970\,\mathrm{mW\,m^{-2}}$ plume, which agrees with the observed ice surface for the region (Scambos and Haran, 2002).

Hydrology parameters are unfortunately highly uncertain, and different choices would lead to a more or less responsive hydrological system and hence possibly a lower GHF value to sustain the fast flow. However, we have a rather low transmissivity of the inefficient drainage system, resulting in low efficiency in water evacuation, causing our system to be sensitive to an increase in water input. If the transmissivity was lowered further, the efficient drainage system is likely to activate in the GHF anomaly region, lowering the water pressure and becoming less sensitive to increased water input. For this reason, we do not expect that a different hydrology configuration would reproduce NEGIS with a lower heat flux. In addition, the subglacial hydrology is only one-way coupled to ice dynamics, so we do not capture the positive feedback expected with higher velocities leading to more melt, and lower effective pressure, giving even higher velocities. With a more responsive and fully coupled system, one might be able to reproduce NEGIS with lower heat flux.

With a simple bed elevation dependent friction and hydrology model forced by melt rates from GHF, we capture the overall pattern of NEGIS velocity. This has implications for studies trying to predict the response of NEGIS to a future climatic warming. Basal friction may not remain constant in time, and thus we cannot fully rely on inversion as it masks unknown time varying basal properties. By using our approach (with or without the GHF anomaly) one can capture complex velocity patterns, and then invert for the remaining basal properties. These may in turn be assumed to be constant in time, while the subglacial hydrology will evolve with a changing climate accounting for varying basal conditions. Unfortunately, observations and estimates of GHF and subglacial hydrology are challenged by large uncertainties. Therefore, it is critical for future observational and modelling studies, to better constrain the basal conditions of the Greenland Ice Sheet.

## 5 Conclusions

Present day basal melt rates from GHF maps and frictional heat are not sufficient to sustain the observed upstream velocities of the Northeast Greenland Ice Stream (NEGIS). The downstream velocities appear to be driven by topography and the spatial pattern is well captured by the subglacial hydrology model. Our findings suggest that a local heat flux anomaly may explain the characteristic high upstream velocity of NEGIS, and hence is consistent with previous studies (Fahnestock et al., 2001; Macgregor et al., 2016; Alley et al., 2019). To reproduce high upstream velocities at the onset of NEGIS, a sustained basal melt rate of $100\,\mathrm{mm\,yr^{-1}}$ is needed in a local region close to EGRIP, where observed present day velocities reach $50\,\mathrm{m\,yr^{-1}}$. Hence, the minimal heat flux value needed to initiate the ice stream in our model is $970\,\mathrm{mW\,m^{-2}}$, as proposed by Fahnestock et al. (2001). This magnitude is too high to be explained by GHF alone, and we suggest that processes such as hydrothermal

circulation may locally elevate the heat flux of the area.

*Code and data availability.* ISSM software is open source and can be downloaded at https://issm.jpl.nasa.gov/. The surface mass balance forcing used in this study, from J.E. Box, is available from https://zenodo.org/record/3359192#.XUSmSpNKhR4 (Box, 2019).

*Author contributions.* SSJ designed the study with help from BdF and KHN. SSJ ran the simulations. NS helped greatly setting up the ice flow model, BdF helped setting up the hydrology model, and HS helped setting up the mantle Plume model. SSJ wrote the manuscript with substantial contributions from all co-authors. The research related to the paper was discussed by all co-authors.

*Competing interests.* The authors declare that they have no conflict of interest.

*Acknowledgements.* SSJ, BdF and KHN were funded by the Ice2Ice project that has received funding from the European Research Council under the European Community's Seventh Framework Programme (FP7/2007-2013)/ERC grant agreement No. 610055. BdF is also funded by SWItchDyn NRC grant(287206). Funding for HS and NS were provided by grants from NASA Cryospheric Science and Modeling, Analysis and Prediction (MAP) Programs. We would like to thank the reviewers Nicholas Holschuh and Signe Hillerup Larsen for greatly improving the manuscript. We would also like to thank Irina Rogozhina and Ralf Greve for good discussions and recommendations.

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

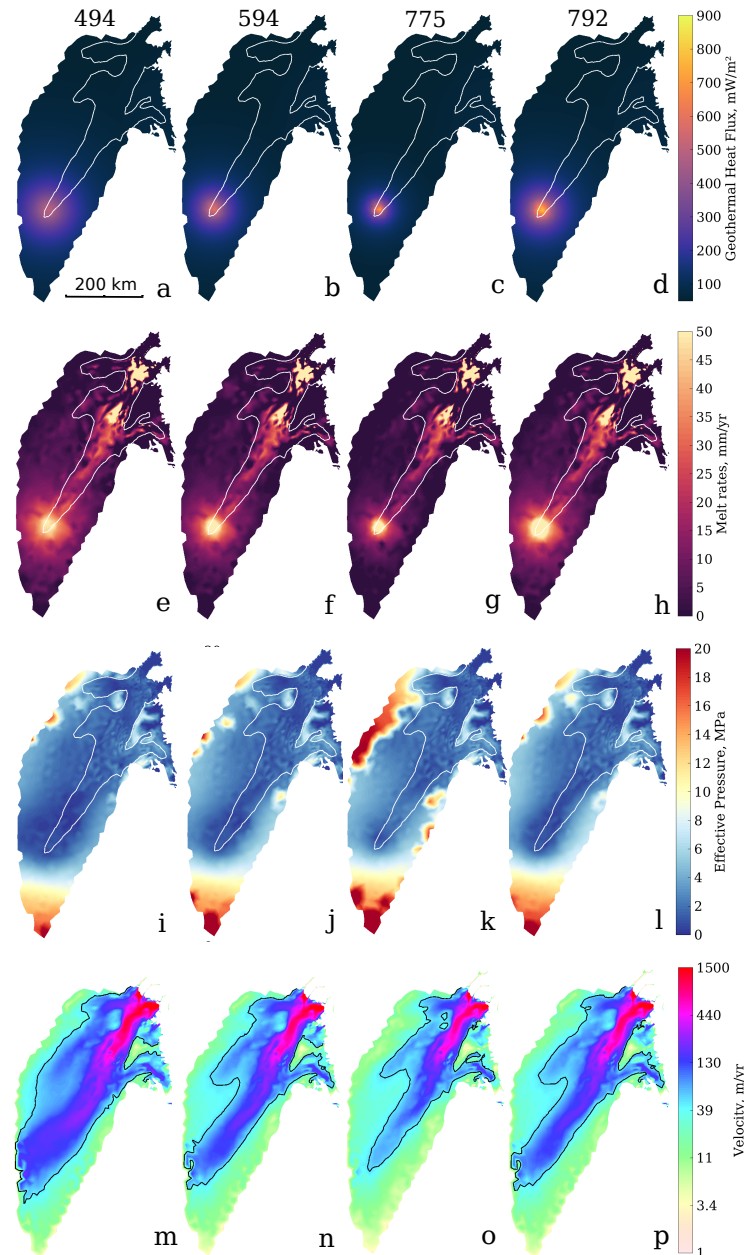

**Figure 4.** Model results from the sensitivity simulations investigating a reduced magnitude and increased size of the mantle plume. First column shows results from 494 Plume with a 300 km radius at 3000 km depth, second column represents 594 Plume with 200 km radius and 2500 km depth, third column represents 775 Plume with 100 km radius and 2000 km depth the last column represents Plume 792 with 200 km radius and 3000 km depth. a-d show the GHF, e-h the resulting basal melt rates, i-l the compute effective pressure and m-p the modelled surface velocity. White lines show the $50\,\mathrm{m\,yr}^{-1}$ observed velocity contour, and black lines show the $50\,\mathrm{m\,yr}^{-1}$ modelled velocity contour.

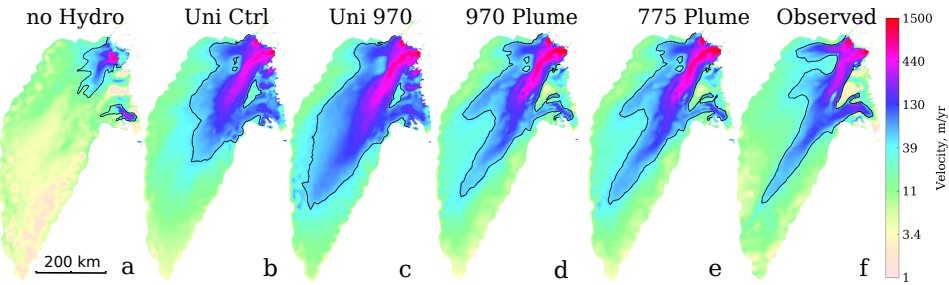

**Figure 5.** Surface velocity results from the no Hydro (a) with effective pressure approximated to the hydrostatic pressure assuming direct connection to the ocean, commonly used in ISSM. Uni Ctrl (b) and plume970-uni experiment (c) use a uniform friction coefficient $\alpha$ set equal to $90\,(\mathrm{ms}^{-1})^{1/2}$. Corresponding GHF, basal melt rates and effective pressure are the same as Ctrl and plume970, shown in Figure 2. For reference we include d and e respectively showing the plume970 and plume 775 simulation (same as Figure 2q, and Figure 4o), and f showing the observed surface velocities interpolated onto the model mesh. Black lines show the $50\,\mathrm{m\,yr}^{-1}$ velocity contour.

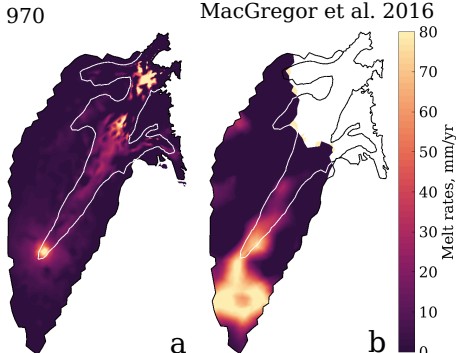

**Figure 6.** Comparison of the basal melt rates computed for plume970 experiment (a) and the gridded basal melt rate estimates of Macgregor et al. (2016) interpolated onto our model mesh (b). White lines show the observed $50\,\mathrm{m\,yr}^{-1}$ velocity contour.