# Peer review of "Exceptionally High Heat Flux Needed to Sustain the Northeast Greenland Ice Stream"

_The Cryosphere, 2019_

## Referee Comment (RC1) · Nicholas Holschuh (Referee) · 29 Oct 2019

**Review of:** *Exceptionally High Geothermal Heat Flux Needed to Sustain the Northeast Greenland Ice Stream*
**Submitted to:** *The Cryosphere Discussions*
**Reviewer:** Nicholas Holschuh

**General Comments:**

This study highlights the role of subglacial hydrology in Northeast Greenland Ice Stream (NEGIS) dynamics. The authors show how both the total melt-water supply and melt-water pathways might explain the overall pattern of ice flow, and provide a compelling case for the use of models that capture dynamic basal hydrology instead of those that define a set of unchanging bed properties from an initial inversion. I believe this paper will make a valuable contribution to both our understanding of NEGIS models, and our projections of future NEGIS behavior.

However, I believe the text needs to more clearly state that this is (at its core) a model sensitivity study. The tone of the paper shifts between making broad physical interpretations (e.g., "Our findings … confirm previous studies [which found a geothermal flux at NEGIS of 1 W/m^2]" -- line 289/90) and making narrower interpretations grounded in the model limitations ("the minimal geothermal heat flux value needed to initiate the ice stream **in our model** is…" -- line 293). I know the authors are aware that each of their statements has an implied caveat (that this is only a model) but it would be easy for the reader to misinterpret the results, and read the presented conclusions as unambiguous constraints on the physical system, given the language used. This includes the title, which I think overstates the conclusions of the paper and should be modified. I have highlighted statements that should be scaled back to more modest claims in the line item comments.

In addition, I am curious about the other output fields of the model. For any model that relies on a substantial basal melt anomaly, I think it is important to show the surface elevation field that is produced. If there is a measurable surface depression at the site of the plume according to the model, that would highlight an important source of disagreement between model and data, as there is no surface depression at the onset of NEGIS. It is likely that the radar methods of Fahnestock and MacGregor overestimate the actual basal melt rates at NEGIS – if similar melt rates applied in this model produce a surface profile much different than the real NEGIS, that must be presented. Regardless, it is impressive that the flow-speed pattern **can** be explained by large volumes of basal melt, but a fuller comparison of model and data will help the reader understand if it **does** explain the flow-speed pattern.

I leave the paper convinced that reproducing basal hydrology is important for reproducing NEGIS in ice sheet models, but the constraints on specific values for heat flux and the area spanned by the anomaly should be taken with a grain of salt.

**Line-Item Corrections:**

| Line #: 10-11 | This statement, in isolation, is too strong. It should include something like "Within our model experiment, a minimum heat flux value … was required to reproduce observed NEGIS velocities." |
| --- | --- |
| Line #: 22 | "information that is needed" |

| Line #: 30-32 | One thing that we found in a modeling study of NEGIS we performed was that the shear margins are likely characterized by a complex velocity and viscosity structure. What did you do for your viscosity initialization in this model? Does it evolve with ice temperature? I am not trying to imply it needs to be cited here, but you may find some of the results from our study interesting and relevant: Holschuh, N., Lilien, D., & Christianson, K. (2019). Thermal Weakening, Convergent Flow, and Vertical Heat Transport in the Northeast Greenland Ice Stream Shear Margins. Geophysical Research Letters, 46, 8184–8193. https://doi.org/https://doi.org/10.1029/2019GL083436 |
|---|---|
| Line #: 40 | Unless there is extraordinary need, you should not cite work in review. It makes it impossible for a reader to evaluate this statement, as it has not been vetted by the peer review process. |
| Line #: 43-44 | Again, I would remove references to papers in review. Without more context, I cannot tell what this sentence means, and I cannot evaluate the claim. What do you mean by uncertainty in the ice flux, our observations of ice thickness and velocity near the grounding-line are quite good? |
| Line #: 47 | This paragraph should include the statement that you make in line 221-224, making very clear to the reader you do not think a mantle plume is presently beneath NEGIS. You are simply using a plume model to generate feasible scenarios that can be tested with the model. Without the sentence at 221, It would be easy to walk away from this paper thinking you believe there is a mantle plume presently under NEGIS (which would require substantially more evidence to justify). |
| Line #: 55 | How was the model changed from Schlegel to the in review paper? If you are including those modifications here, it is important that the reader know what they are, but they cannot be determined as the paper referenced is not published. This is a case where an in review citation may be acceptable, but you need to include the salient details from the paper in the text here. |
| Line #: 58 | Could you provide justification for your choice in sliding law here? |
| Line #: 87-88 | This statement does not agree with the seismic results collected at the onset of NEGIS, where there was no apparent relationship between topography and till strength. You should reference whether or not this argument is observationally substantiated. It would be helpful to include discussion here from Christianson et al: Christianson, K., Peters, L. E., Alley, R. B., Anandakrishnan, S., Jacobel, R. W., Riverman, K. L., … Keisling, B. A. (2014). Dilatant till facilitates ice-stream flow in northeast Greenland. Earth and Planetary Science Letters, 401, 57–69. https://doi.org/10.1016/j.epsl.2014.05.060 |

| | |
|---|---|
| Line #: 101 | The plumes discussed here are not very consistent with MacGregor et al 2016, who find large areas of basal melt (> 100km x 100km) well upstream of NEGIS. I think the agreement between Fahnestock and MacGregor throughout the manuscript is generally overstated. |
| Line #: 137 | Clarify what you mean here, Fahnestock and MacGregor did not have identical results. |
| Line #: 163-164 | Here is an example of potentially misleading language -- you show the elevated heat required by your model to initiate NEGIS. Much less heat may be required if the bed were uniformly weaker, if you included fabric evolution or imposed viscosity transitions, if the water transmissivity at the bed were lower, etc. All of the values you provide are contingent on the physical processes included in the model, the assumptions about the flow law form and parameters, and the experimental design. |
| Line #: 168 | "met" should be "melt" |
| Line #: 173-174 | "This shows that plumes with a restricted extent, ~50km x 50km, produce model results more consistent with the observed flow behavior in the upstream reaches of NEGIS." -- something that clarifies that this is not a necessary condition for NEGIS. |
| Line #: 197-198 | Perhaps change this sentence to read "the geothermal heat flux needed to induce the observed upstream velocity of NEGIS in our model is ~970, consistent with values presented in Fahnestock et al. (2001)." What you are stating here (and in your next sentence) is essentially "high melt water production rates are required to drive fast flow in the upstream region sof NEGIS, assuming the absence of other variations in bed strength driven by substrate heterogeneity". I think that last caveat is important to make here and elsewhere in the paper; you are forcing all of the variation to be driven by hydrology, but it need not be the only property that varies in space. |
| Line #: 211-212 | The comparison with Jarosch and Gudmundsson (2007) here seems odd, as they apply their geothermal flux anomaly over ~500m. No one would argue that their anomaly could exist at the scale of your plume. However, their results do highlight something that I think you should present to your reader -- substantial melt anomalies manifest in the ice sheet surface. I imagine the ice sheet surface in your models has a similar (albeit smaller) melt feature as the one in Jarosch and Gudmundsson. If so, somewhere in this work you should state that localized, substantial melt under NEGIS would be visible at the ice sheet surface, but is not apparent in altimetry data. Any discrepancy (or, if present, agreement) in the effect of basal melt on the ice surface profile must be discussed. |

| Line #: 218-219 | This seems to imply that your results differ because you are fitting to velocities instead of temperatures, but that is not the primary factor. Greve has no constraints near the onset of NEGIS, while your study does. If the anomaly you argue for existed, Greve would have no way of knowing with the data he has available. Greve's data set is actually a much more direct measure of geothermal flux -- if he had broader observational coverage it would be hard to argue with his results. |
|---|---|
| Line #: 221-223 | As stated earlier, this sentence should come much sooner in the paper. Without additional data, we have no means of explaining why there might be a heat flux anomaly at NEGIS, and it is not likely a modern plume. |
| Line #: 227-228 | MacGregor et al. have abnormally high melt rates in several places in Greenland, including over a broad region upstream of NEGIS and in SW Greenland. This citation here seems inconsistent with the statement made. |
| Line #: 273-277 | A broader discussion of the role of the friction law would be useful. What if you used a non-linear sliding law? What direction would that change your results? It would be useful for the reader to understand how the plume characteristics you describe would need to vary to reproduce NEGIS using a range of different model set-ups. |
| Line #: 290 | "confirms previous studies" is too strong. "is consistent with" would be better. |

---

## Referee Comment (RC2) · Signe Hillerup Larsen (Referee) · 7 Nov 2019

General comments:

The paper builds on the hypothesis, put forward by Fahnstock (2001) amongst others, of the existence of a geothermal heat anomaly at the initiation of the North East Greenland Ice Stream (NEGIS). The Ice Sheet System Model is used as a tool to test this hypothesis. The model experiments presented are relevant and rigorous, while also building on a large modelling effort by the co-authors in previous publications.

The paper is in most parts clearly written and model setup is well described. The

discussion is thorough on the topic of how basal meltwater will affect ice flow patterns. Results and conclusion are convincing in a sense that the study makes a good case for a strong geothermal heat flux anomaly to be the reason for NEGIS to originate to far inland.

In discussing the results I am however missing some comments on uncertainties in ice flow viscosity which have been shown by a few studies (Van der Veen (2011) - Controls on the recent speed-up of Jakobshavn Isbræ, West Greenland; and Bondzio et al. (2017) - The mechanisms behind Jakobshavn Isbrae's acceleration and mass loss a 3D thermomechanical model study) to be important for maintaining ice stream flow. Se specific comments on the paper content below.

The presentation of the experiments and results could also be improved as written in details in the specific comments on the structure of the paper.

Specific comments on the structure of the paper:

1. Structure of method and result section: a) The storyline in the experiment and result section does not match. In the results section the focus is on the study testing the hypothesis of the existence of a geothermal heat flux anomaly of 970 mW/m2. The rest of the experiments are described as sensitivity studies to this main hypothesis. This is not the story line in the experiment section.

2. Results section: a) Presentation of results: I think it's a good idea to use the 50 m/yr contour to compare results. Maybe add some meta text in the beginning explaining that this is your approach and if possible add the observed contour line on all result plots for comparison?

b) In the first paragraph of the results section the Ctrl simulation is described as a way to obtain the basal melt rate, and then in the same paragraph the resulting velocity field is explained. I find this a bit confusing. Maybe just stick to the explanation about the velocity field, because the method to obtain N is already described in the methods

section.

Specific comments on content:

3. Discussion section: a) the discussion is purely focussed on the ice/bed interface, but I am wondering about how the resulting flow pattern depends on uncertainties within the ice such as viscosity and the fact that shear margins are not resolved by the 15 km grid. Thus a short discussion of ice viscosity, shear margins and model resolution should in my opinion be included.

b) The aim is to have a model that is independent of present day observations. This is not strictly met in the way N is obtained, which is clearly explained. However, the bedmap is also based on modelling using present day velocity observations, which could bias the results, this makes the basal friction coefficient relate to observed velocities in a more diffuse way. This should also be mentioned somewhere.

4. Conclusions: a) Conclusions appear a bit too conclusive, and the authors should make an effort to make it clearer that they are aware that this is a relatively simple test of the hypothesis that a geothermal heat flux anomaly could explain the onset of NEGIS.

Line by line comments:

60-65: Effective pressure is defined in words twice.

153-154: The last sentence of the paragraph makes it sound a bit like that the 970mW/m2 experiment represents reality. Maybe just explain how the ice stream signature becomes weaker with lower forcing.

199: I am wondering if the width of the modelled ice stream could be related to model underestimating viscosity?

212-213: The sentence starting with: 970 mW/m2 is only... should be moved to methods section

222: Maybe refer to Martos et al, 2018 or other paper that describes the continental passage over the Icelandic hotspot. This information should probably be included in the introduction or methods section.

281: By inverting for basal friction you not only create a basal friction map that cannot evolve in time, you also place all uncertainty from the model viscosity for example in the basal friction map.

Figures:

Figure 1: Include the place names used in the text e.g. Storstrømmen and Zachariæ

Figure 2, 3, 4 and 5: Maybe show the observed (white) 50m/yr contour in all the velocity plots where only the modelled contour is shown.

References: The reference to the Fox Maule paper or data is incomplete.

---

## Author Comment (AC1) · 20 Dec 2019

**Response to Reviewers**

December 20, 2019

**General response**

Our thanks to Nicholas Holschuh and Signe Hillerup Larsen, their suggestions greatly improved the manuscript. We are grateful for their insights in both observations and modelling of NEGIS. The major changes to the manuscript are outlined here:

- We moderated our language in our statements and included an extended caveat section in the discussion. We also modified the abstract to underline that we present model results, and the conclusion now includes model caveats and uncertainties. With a more detailed abstract and conclusion including model assumptions, we avoid being too assertive, allowing the reader to think critically about the numbers presented. For these reasons, we choose to keep the title as is.

- Following your suggestions, we have provided more details on the comparisons to previous studies throughout the manuscript. In addition, we included a figure of the gridded melt dataset from Macgregor et al. (2016), interpolated onto our model mesh. This allows for an improved and more direct comparison. We also carefully restructured the section comparing our geothermal heat flux results to previous findings in the discussion, to clearly state how high the implied value presented here would be.

- We included a new equation on the viscosity

- Following recommendations from Irina Rogozhina during Smith-Johnsens PhD defence and discussion thereafter, we have decided to make some additional changes regarding terminology. We explain our high values (970 mW/m$^2$) by advective heat transport (hydrothermal circulation). The term "geothermal heat flux" is inaccurate, as it only comprises pure conductive heat transfer. We removed 'geothermal' from the title, and expanded the abstract, discussion and conclusion to include this.

On behalf of the authors,
Silje Smith-Johnsen (PhD)

**RC1: Nicholas Holschuh**

**Comment**

In addition, I am curious about the other output fields of the model. For any model that relies on a substantial basal melt anomaly, I think it is important to show the surface elevation field that is produced. If there is a measurable surface depression at the site of the plume according to the model, that would highlight an important source of disagreement between model and data, as there is no surface depression at the onset of NEGIS. It is likely that the radar methods of Fahnestock and MacGregor overestimate the actual basal melt rates at NEGIS –if similar melt rates applied in this model produce a surface profile much different than the real NEGIS, that must be presented. Regardless, it is impressive that the flow-speed pattern can be explained by large volumes of basal melt, but a fuller comparison of model and data will help the reader understand if it does explain the flow-speed pattern.

**Response**

We agree that this is an interesting point to investigate and we looked at the model surface of Ctrl and plume970 and compared them to observations (Scambos & Haran, 2002). We found that our Ctrl simulation underestimates the surface elevation over the model domain. To disentangle the surface lowering caused directly by introducing the plume, we investigated surface differences between the plume970 and the Ctrl simulation. We found that there is no evidence of a local surface depression above the plume. We do observe a regional surface lowering of the entire domain upstream of the plume, and a slight thickening downstream (and hence the surface becomes closer to the observed surface in the downstream area). The lack of a local surface signal from the plume in our model thus agrees with observations.

These findings are in line with previous idealized experiments we have conducted where we found that the plume melts the above lying ice creating a local depression, if no subglacial hydrology model is included. However if one includes a subglacial hydrology model like this study, the ice is allowed to slide in response to the extra water added at the base. The surface signal of the plume becomes more regionally dispersed due to both effective pressure changes and most importantly the advection of ice downstream distributes the surface signal. We included a statement of the surface results in L293-295.
* * *
**Comment**

Line#: 10-11
This statement, in isolation, is too strong. It should include something like "Within our model experiment, a minimum heat flux value ... was required to reproduce observed NEGIS velocities.

**Response**

Thanks, done
* * *
**Comment**

Line #: 22
"information that is needed"

**Response**

Thanks, done
* * *
**Comment**

Line #: 30-32

One thing that we found in a modeling study of NEGIS we performed was that the shear margins are likely characterized by a complex velocity and viscosity structure. What did you do for your viscosity initialization in this model? Does it evolve with ice temperature? I am not trying to imply it needs to be cited here, but you may find some of the results from our study interesting and relevant: Holschuh, N., Lilien, D., and Christianson, K. (2019). Thermal Weakening, Convergent Flow, and Vertical Heat Transport in the Northeast Greenland Ice Stream Shear Margins. Geophysical Research Letters, 46, 8184–8193. https://doi.org/https://doi.org/10.1029/2019GL083436

**Response**

Yes, ice viscosity is temperature dependant, and evolves through time with changes in temperature. This is now included in equation 2 in the ice sheet model description, L73. For temperature we initialize with prescribed surface temperatures and the basal boundary condition, and we solve a thermal steady state. No climatological spin-up is used, and therefore the overall thermal state may be too warm. We also use the 3D Higher-Order approximation to compute vertical velocities, important in the shear margins. As stated by the reviewer Signe Hillerup Larsen, we have a rather coarse mesh in the shear margins and may thus underestimate the strain heating that is occurring here. We included a caveat section on how this may influence our results in the discussion (L276-279).
* * *
**Comment**

Line#: 40

Unless there is extraordinary need, you should not cite work in review. It makes it impossible for a reader to evaluate this statement, as it has not been vetted by the peer review process.

**Response**

Thank you, we removed the Smith-Johnsen et al. A as this is still in review. We chose to keep the Smith-Johnsen et al. B, as this manuscript is now accepted (Smith-Johnsen, Schlegel, de Fleurian & Nisancioglu, accepted).
* * *
**Comment**

Line#: 43-44

Again, I would remove references to papers in review. Without more context, I cannot tell what this sentence means, and I cannot evaluate the claim. What do you mean by uncertainty in the ice flux, our observations of ice thickness and velocity near the grounding-line are quite good?

**Response**

In this paper we show how uncertainties in model inputs (GHF) propagate through the ice flow model and cause a large range of modelled mass (ice) flux through NEGIS, and therefore large uncertainties. This is relevant for future predictions, as we do not have observations. We chose to keep the Smith-Johnsen et al. B, as this manuscript is now accepted (Smith-Johnsen et al., accepted).
* * *
**Comment**

Line#: 47

This paragraph should include the statement that you make in line 221-224, making very clear to the reader you do not think a mantle plume is presently beneath NEGIS. You are simply using a plume model to generate feasible scenarios that can be tested with the model.Without the sentence at 221, It would be easy to walk away from this paper thinking you believe there is a mantle plume presently under NEGIS (which would require substantially more evidence to justify).

**Response**

Thank you this is a good point. We have edited the paragraph accordingly by including this statement earlier in the paper, L50, and removed it from the discussion.
* * *
**Comment**

Line#: 55

How was the model changed from Schlegel to the in review paper? If you are including those modifications here, it is important that the reader know what they are, but they cannot be determined as the paper referenced is not published. This is a case where an in review citation may be acceptable, but you need to include the salient details from the paper in the text here.

**Response**

Thank you for pointing this out to us. The most important change is that the thermomechanical ice flow model is coupled to a subglacial hydrology model. We changed the sentence to include this, and keep the reference to Smith-Johnsen et al. (accepted), as it is now accepted.
* * *
**Comment**

Line#: 58

Could you provide justification for your choice in sliding law here?

**Response**

This is the most commonly used sliding law in ISSM. It was used by (Schlegel, Larour, Seroussi, Morlighem & Box, 2015) and (Smith-Johnsen et al., accepted), so to avoid a complete new model set-up with following spin-up, we decided to keep it. Instead of justifying the choice of sliding law here, we included a discussion on the implications of using this in the new caveat section of the discussion (L269-275).
* * *
**Comment**

Line#: 87-88

This statement does not agree with the seismic results collected at the onset of NEGIS, where there was no apparent relationship between topography and till strength. You should reference whether or not this argument is observationally substantiated. It would be helpful to include discussion here from Christianson et al: Christianson, K., Peters, L. E., Alley, R. B., Anandakrishnan, S., Jacobel, R. W., Riverman, K. L., ... Keisling, B. A. (2014). Dilatant till facilitates ice-stream flow in northeast Greenland. Earth and Planetary Science Letters, 401, 57–69. https://doi.org/10.1016/j.epsl.2014.05.060

**Response**

Thank you for this reference. We do compare our friction coefficient distribution to roughness observations in L97-98. The friction coefficient includes everything unknown at the bed, in addition to till strength. We are aware of the limitations of using this simple approach for the friction coefficient, and we therefore mention these limitations in the new caveat section in the discussion.
* * *
**Comment**

Line#: 101

The plumes discussed here are not very consistent with MacGregor et al 2016, who find large areas of basal melt (> 100km x 100km) well upstream of NEGIS. I think the agreement between Fahnestock and MacGregor throughout the manuscript is generally overstated.

**Response**

We tried to explain that the plumes here compares well to the northeastern branch of the anomaly of Macgregor et al. (2016). We have made this clearer by removing the references to figures within their paper, and instead included a plot of the Macgregor et al. (2016) gridded dataset for our model domain in a new Figure 6. This improves the understanding of the reader, and allow for a more direct comparison of the basal melt rates from our 970plume experiment to the dataset.

Overall in the paper we have modified the comparison of the GHF and basal melt to previous studies, by providing more details for each comparison.
* * *
**Comment**

Line#: 137

Clarify what you mean here, Fahnestock and MacGregor did not have identical results.

**Response**

Thank you, we mean the maximum magnitude of geothermal heat flux (970 $mW/m^2$) proposed by Fahnestock, Abdalati, Joughin, Brozena and Gogineni (2001). We have clarified this, and we removed the Macgregor et al. (2016) reference.
* * *
**Comment**

Line#: 163-164

Here is an example of potentially misleading language –you show the elevated heat required by your model to initiate NEGIS. Much less heat may be required if the bed were uniformly weaker, if you included fabric evolution or imposed viscosity transitions, if the water transmissivity at the bed were lower, etc. All of the values you provide are contingent on the physical processes included in the model, the assumptions about the flow law form and parameters, and the experimental design.

**Response**

Thank you, we toned down the statement by writing "indicate" in stead of "show", and we included "in our model". In addition, we have included a caveat section in the discussion where we provide several reasons for why we may overestimate the geothermal heat of the plume, due to model uncertainties and assumptions (friction law, shear margin softening, subglacial hydrology parameters).

However, if the bed were uniformly weaker the entire domain would speed up, resulting in an even lower surface, and increase the underestimation of ice thickness. In addition, the outlet glaciers are too fast in our model, and a uniformly weaker bed would intensify the problem. We agree that this is a very simple estimate of basal friction, and we also included this in the caveat section in the discussion.
* * *
**Comment**

Line#: 168

"met" should be "melt"

**Response**

Thanks, done
* * *
**Comment**

Line#: 173-174

"This shows that plumes with a restricted extent, 50km x 50km, produce model results more consistent with the observed flow behavior in the upstream reaches of NEGIS." –something that clarifies that this is not a necessary condition for NEGIS.

**Response**

Thanks, done
* * *
**Comment**

Line#: 197-198

Perhaps change this sentence to read "the geothermal heat flux needed to induce the observed upstream velocity of NEGIS in our model is 970, consistent with values presented in Fahnestock et al. (2001)." What you are stating here (and in your next sentence) is essentially "high melt water production rates are required to drive fast flow in the upstream regions of NEGIS, assuming the absence of other variations in bed strength driven by substrate heterogeneity". I think that last caveat is important to make here and elsewhere in the paper; you are forcing all of the variation to be driven by hydrology, but it need not be the only property that varies in space.

**Response**

We agree, and changed the sentence to what you suggested. However, the next statement is not exactly true, as our friction coefficient is spatially varying and not uniform, and represents everything that varies at the base. We show the importance of spatially varying bed properties by running two simulations where we have a spatially uniform friction coefficient (simulation "Uni Ctrl" and "Uni 970") where the velocity pattern is less confined and less similar to observations.

We agree that it is important to state that we keep everything constant in our model and only vary GHF, and try to explain the observed velocities by subglacial hydrology, despite the many model assumptions and uncertainties. We stated this at the beginning of the new caveat section in the discussion, L267.
* * *
**Comment**

Line#: 211-212

The comparison with Jarosch and Gudmundsson (2007) here seems odd, as they apply their geothermal flux anomaly over 500m. No one would argue that their anomaly could exist at the scale of your plume. However, their results do highlight something that I think you should present to your reader –substantial melt anomalies manifest in the ice sheet surface. I imagine the ice sheet surface in your models has a similar (albeit smaller)melt features the one in Jarosch and Gudmundsson. If so, somewhere in this work you should state that localized, substantial melt under NEGIS would be visible at the ice sheet surface, but is not apparent in altimetry data. Any discrepancy (or, if present, agreement) in the effect of basal melt on the ice surface profile must be discussed.

**Response**

Thank you, we agree and have removed this reference. In addition, Iceland is generally not a representative comparison as it is located on a mantle plume and a spreading ridge, thus an extreme geothermal heat flux example. This is a very interesting topic. As mentioned earlier, by comparing the plume970 simulation to the Ctrl, to disentangle the direct impact of the plume, we do not see a significant local surface depression. We think this is due to the hydrology dispersing the signal, and most importantly the advection of ice redistributes and dampens the surface signal. However, we think the case would be different if the ice above the local plume was not sliding. It would be interesting to test our plume in an area where a local mantle plume would not trigger fast flow, only local melt, to see if the surface expression would look different.
* * *
**Comment**

Line#: 218-219

This seems to imply that your results differ because you are fitting to velocities instead of temperatures, but that is not the primary factor. Greve has no constraints near the onset of NEGIS, while your study does. If the anomaly you argue for existed, Greve would have no way of knowing with the data he has available. Greve's data set is actually a much more direct measure of geothermal flux–if he had broader observational coverage it would be hard to argue with his results.

**Response**

This is true and a good point, and we removed this reasoning from the paper. We compare our GHF values to the highest estimate of Greve, and clearly state that this is from NGRIP. We have restructured this entire section and the following section to be more reader friendly.
* * *
**Comment**

Line#: 221-223

As stated earlier, this sentence should come much sooner in the paper. Without additional data, we have no means of explaining why there might be a heat flux anomaly at NEGIS, and it is not likely a modern plume.

**Response**

Thank you, as indicated earlier, we moved this sentence to the introduction L50.
* * *
**Comment**

Line#: 227-228
MacGregor et al. have abnormally high melt rates in several places in Greenland, including over a broad region upstream of NEGIS and in SW Greenland. This citation here seems inconsistent with the statement made.

**Response**

Thank you, we removed the statement as it did not contribute to the section. We did, as mentioned, include a plot of the melt anomaly by Macgregor et al. (2016) (Fig 6) for an improved, direct comparison of both magnitude and spatial extent.
* * *
**Comment**

Line#: 273-277
A broader discussion of the role of the friction law would be useful. What if you used a non-linear sliding law? What direction would that change your results? It would be useful for the reader to understand how the plume characteristics you describe would need to vary to reproduce NEGIS using arange of different model set-ups.

**Response**

Yes this is a caveat of our model set-up, and based on your recommendation we extended the discussion of the linear friction law in the new caveat section of the discussion (L269-275). We agree that the plume would change given a different model setup, and this is discussed in more detail in the section starting at L267.
* * *
**Comment**

Line#: 290
"confirms previous studies" is too strong. "is consistent with" would be better

**Response**

Thank you, we changed this to your suggestion.
* * *
**RC2: Signe Hillerup Larsen**

**Comment**

1. Structure of method and result section: a) The storyline in the experiment and result section does not match. In the results section the focus is on the study testing the hypothesis of the existence of a geothermal heat flux anomaly of 970 mW/m2. The rest of the experiments are described as sensitivity studies to this main hypothesis. This is not the story line in the experiment section.

**Response**

Thank you for noticing this, we have changed the storyline in the experiment section to match the one in the results. More specifically we removed the range of GHF in the sensitivity studies in the beginning of the experiment section. The storyline in the experiment section is now the following: first we present the 970 plume experiment; explain why we need a Ctrl; and finally we present the sensitivity simulations and explain their purpose. In the results we start with the Ctrl in order to explain the background values for all the simulations.
* * *
**Comment**

2. Results section: a) Presentation of results: I think it's a good idea to use the 50 m/yr contour to compare results. Maybe add some meta text in the beginning explaining that this is your approach and if possible add the observed contour line on all result plots for comparison?

**Response**

Good suggestion, we added a description of how we evaluate the performance of each model simulation using these contours in L138-140. We agree, and originally tried to include both modeled and observed velocity contour in the results plot. However it was messy and too much information in one plot. We therefore decided to show the observed velocity contour on all the result figures apart from the velocity figures where we plotted the modeled velocity contour.
* * *
**Comment**

b) In the first paragraph of the results section the Ctrl simulation is described as a way to obtain the basal melt rate, and then in the same paragraph the resulting velocity field is explained. I find this a bit confusing. Maybe just stick to the explanation about the velocity field, because the method to obtain N is already described in the methods section.

**Response**

Thank you expressing this, we removed the methods part. In fact, we removed all the part of this section concerning methods to avoid unnecessary repetition.
* * *
**Comment**

3. Discussion section:
a) the discussion is purely focussed on the ice/bed interface, but I am wondering about how the resulting flow pattern depends on uncertainties within the ice such as viscosity and the fact that shear margins are not resolved by the 15km grid. Thus a short discussion of ice viscosity, shear margins and model resolution should in my opinion be included.

**Response**

Thank you, this is very good point that we did not include originally. We added a caveat section where we discuss how we could obtain similar high velocity as in the 970 experiment, by changing other parameters in the model and then getting away with lower geothermal heat flux values. In L276-280 we discuss the softening of shear margins and how we may overestimate the lateral drag.
* * *
**Comment**

b) The aim is to have a model that is independent of present day observations. This is not strictly met in the way N is obtained, which is clearly explained. However, the bedmap is also based on modelling using present day velocity observations, which could bias the results, this makes the basal friction coefficient relate to observed velocities in a more diffuse way. This should also be mentioned somewhere.

**Response**

Thank you, we agree. We included this caveat in L289-290.
* * *
**Comment**

4. Conclusions: a) Conclusions appear a bit too conclusive, and the authors should make an effort to make it clearer that they are aware that this is a relatively simple test of the hypothesis that a geothermal heat flux anomaly could explain the onset of NEGIS.

**Response**

We modified the conclusion and added a sentence on model caveats, allowing the reader to understand how the number presented is dependent on model uncertainties (L323). As explained above we added a section in the discussion where we suggest other ways we could trigger fast flow of NEGIS in our model, apart from the geothermal heat flux.
* * *
**Comment**

Line#: 60-65
Effective pressure is defined in words twice.

**Response**

Thank you, we fixed that.
* * *
**Comment**

Line#: 153-154
The last sentence of the paragraph makes it sound a bit like that the 970 $mW/m^2$ experiment represents reality. Maybe just explain how the ice stream signature becomes weaker with lower forcing.

**Response**

Thanks, we toned down and included 'given our model set-up' in this statement.
* * *
**Comment**

> Line#: 199
>
> I am wondering if the width of the modelled ice stream could be related to model underestimating viscosity?

**Response**

This is a very good point, and may explain the more diffuse modelled velocity pattern and lack of sharp gradients in the shear margins. We added your suggestion about width in the shear margin viscosity discussion, L279-280, thanks.
* * *
**Comment**

> Line#: 212-213
>
> The sentence starting with: 970 $mW/m^2$ is only...should be moved to methods section.

**Response**

We agree that it is too late to include here. We find it more a result than a method, as this is computed by the plume model and not prescribed. We removed this statement from the discussion, as it is not important. We generally restructured the section in the discussion where we compare our findings to previous studies, and try to better explain why our values are so high.
* * *
**Comment**

> Line#: 222
>
> Maybe refer to Martos et al, 2018 or other paper that describes the continental passage over the Icelandic hotspot. This information should probably be included in the introduction or methods section.

**Response**

Thank you, we agree and we moved this statement to the introduction (L50). And for the high background geothermal heat flux due to Iceland plume we refer to Rogozhina et al. 2016 and Martos et al. 2018 (L38).
* * *
**Comment**

> Line#: 281 By inverting for basal friction you not only create a basal friction map that cannot evolve in time, you also place all uncertainty from the model viscosity for example in the basal friction map.

**Response**

Yes this is true, everything uncertain in the model is blamed on the spatially varying 'bed properties'.
* * *
**Comment**

> Figure 1:
>
> Include the place names used in the text e.g. Storstrømmen and Zachariæ.

**Response**

Great suggestion, we included this in Figure 1c, where we introduce EGRIP and the model domain.

**Comment**

Figure 2, 3, 4 and 5:

Maybe show the observed (white) 50 m/yr contour in all the velocity plots where only the modelled contour is shown.

**Response**

As stated above, we originally tried this, but the figure was not clear so we avoided this.
* * *
**Comment**

References:

The reference to the Fox Maule paper or data is incomplete.

**Response**

Well spotted, we completed this reference, thank you.
* * *
**References**

Fahnestock, M., Abdalati, W., Joughin, I., Brozena, J. & Gogineni, P. (2001, dec). High Geothermal Heat Flow, Basal Melt, and the Origin of Rapid Ice Flow in Central Greenland. *Science*, *294*(5550), 2338–2342.

Macgregor, J., Fahnestock, M., Catania, G., Aschwanden, A., Clow, G., Colgan, W., ... Seroussi, H. (2016). A synthesis of the basal thermal state of the Greenland Ice Sheet. *Journal of Geophysical Research: Earth Surface*(121), 1328–1350. doi: 10.1002/2015JF003803

Scambos, T. A. & Haran, T. (2002). An image-enhanced DEM of the Greenland ice sheet. *Annals of Glaciology*, *34*, 291–298. doi: 10.3189/172756402781817969

Schlegel, N.-J., Larour, E., Seroussi, H., Morlighem, M. & Box, J. E. (2015). Ice discharge uncertainties in Northeast Greenland from climate forcing and boundary conditions of an ice flow model. *Journal of Geophysical Research: Earth Surface*, *submitted*, 1–21. doi: 10.1002/2014JF003359.Received

Smith-Johnsen, S., Schlegel, N.-J., de Fleurian, B. & Nisancioglu, K. (accepted). Sensitivity of the Northeast Greenland Ice Stream to Geothermal Heat. *Journal of Geophysical Research: Earth Surface*. doi: 10.1029/2019JF005252

---

## Author Comment (AC2) · 20 Dec 2019

Thank you to Nicholas Holschuh and Signe Hillerup Larsen for their helpful reviews. We address their comments in the supplement.

Please also note the supplement to this comment: https://www.the-cryosphere-discuss.net/tc-2019-212/tc-2019-212-AC2-supplement.pdf

---

## Author Response (AR1)

**Response to Reviewers**

December X, 2019

**General response**

Our thanks to Nicholas Holschuh and Signe Hillerup Larsen, your suggestions greatly improved the manuscript. We are grateful for your insights in both observations and modelling of NEGIS. The major changes to the manuscript are outlined here:

- We scaled down the misleading language in our statements, included an extended caveat section in the discussion. We also modified the abstract to underlining that this is model results, and the conclusion now includes model caveats and uncertainties. With a more detailed abstract and conclusion including model assumptions, we avoid being too conclusive, and allowing the reader to think critically about the numbers presented For this reason, we would like to keep the title as is.

- We have provided more detail on the comparisons to previous studies throughout the manuscript, following your suggestions. In addition, we included a figure of the gridded melt dataset from MacGregor et al. 2016, interpolated onto our model mesh. This allows for an improved an more direct comparison, as this was unclear in our previous manuscript. We also carefully restructured the section comparing our geothermal heat flux results to previous findings in the discussion, to clearly state how unrealistic high the value presented here is.

- We included a new equation on the viscosity

- Following recommendations from Irina Rogozhina during Smith-Johnsens PhD defence and discussion therein, we have decided to make some additional changes regarding terminology. We explain our high values (970 mW/m$^2$) by advective heat transport (hydrothermal circulation) and the term "geothermal heat flux" is incorrect, as it only comprises pure conductive heat transfer. We removed 'geothermal' from the title, and expanded the abstract, discussion on and conclusion to include this.

On behalf of the authors,
Silje Smith-Johnsen (PhD)

**RC1: Nicholas Holschuh**

**Comment**

In addition, I am curious about the other output fields of the model. For any model that relies on a substantial basal melt anomaly, I think it is important to show the surface elevation field that is produced. If there is a measurable surface depression at the site of the plume according to the model, that would highlight an important source of disagreement between model and data, as there is no surface depression at the onset of NEGIS. It is likely that the radar methods of Fahnestock and MacGregor overestimate the actual basal melt rates at NEGIS –if similar melt rates applied in this model produce a surface profile much different than the real NEGIS, that must be presented. Regardless, it is impressive that the flow-speed pattern can be explained by large volumes of basal melt, but a fuller comparison of model and data will help the reader understand if it does explain the flow-speed pattern.

**Response**

We agree that this is an interesting point to investigate and we looked at the model surface of Ctrl and plume970 and compared to observations (Scambos and Haram 2002). We found that our Ctrl simulation underestimates the surface elevation over the model domain. To disentangle the surface lowering caused directly by introducing the plume, we investigated surface differences between the plume970 and the Ctrl simulation. We found that there is no evidence of a local surface depression above the plume. We do observe a regional surface lowering of the entire domain upstream of the plume, and a slight thickening downstream (and hence the surface becomes closer to the observed surface in the downstream area).

These findings are in line with previous idealized experiments we have conducted where we found that the plume melts the above lying ice creating an local depression, if no subglacial hydrology model is included. However if one do include a subglacial hydrology model like this study, the ice is allowed to slides in response to the extra water added at the base. the surface signal of the plume becomes more regionally disperse due to both effective pressure changes and most importantly the advection of ice downstream redistribute the surface signal. To investigate if this is the case for a NEGIS model too, we launched a simulation with effective pressure from the Ctrl and the geothermal heat and thus basal melt rates from the plume970. In this set-up the ice dynamics only responds due to thermal changes, not basal sliding. The resulting surface expression above the plume display a deep local depression, and is this strikingly different from our 970plume experiment in the manuscript.

The lack a surface signal from the local plume in our model thus agrees with observations. In addition we would like to state that our basal melt rate estimate ( 100 mm/yr) for the plume970, falls between the values from preliminary radar estimates for EastGRIP presented at the NEGIS consortium meeting in Copenhagen (!!waiting for numbers from Dorthe and Angelika!!)
* * *
**Comment**

Line#: 10-11
This statement, in isolation, is too strong. It should include something like "Within our model experiment, a minimum heat flux value ... was required to reproduce observed NEGIS velocities.

**Response**

Thanks, done
* * *
**Comment**

Line #: 22

"information that is needed"

**Response**

Thanks, done
* * *
**Comment**

Line #: 30-32

One thing that we found in a modeling study of NEGIS we performed was that the shear margins are likely characterized by a complex velocity and viscosity structure. What did you do for your viscosity initialization in this model? Does it evolve with ice temperature? I am not trying to imply it needs to be cited here, but you may find some of the results from our study interesting and relevant: Holschuh, N., Lilien, D., and Christianson, K. (2019). Thermal Weakening, Convergent Flow, and Vertical Heat Transport in the Northeast Greenland Ice Stream Shear Margins. Geophysical Research Letters, 46, 8184–8193. https://doi.org/https://doi.org/10.1029/2019GL083436

**Response**

Yes, ice viscosity is temperature dependant, and evolves trough time with changes in temperature. This is now included in equation 2 in the ice sheet model description, LXX. For temperature we initialize with prescribed surface temperatures, and the basal boundary condition, we run a thermal steady state. No climatological spin-up is used, and therefore the overall thermal state may be too warm. We also use the 3D Higher-Order approximation to compute vertical velocities, important in the shear margins. As stated by the reviewer Signe Hillerup Larsen, we have a rather coarse mesh in the shear margins an may thus underestimate the strain heating that occuring here. We included a caveat section on how this may influence our results in the discussion (LXX).
* * *
**Comment**

Line#: 40

Unless there is extraordinary need, you should not cite work in review. It makes it impossible for a reader to evaluate this statement, as it has not been vetted by the peer review process.

**Response**

Thank you, we removed the Smith-Johnsen et al. A as this is still in review. We chose to keep the Smith-Johnsen et al. B, as it manuscript is now accepted.
* * *
**Comment**

Line#: 43-44

Again, I would remove references to papers in review. Without more context, I cannot tell what this sentence means, and I cannot evaluate the claim. What do you mean by uncertainty in the ice flux, our observations of ice thickness and velocity near the grounding-line are quite good?

**Response**

In this paper we show how uncertainties in model inputs (GHF) propagate through the ice flow model and cause a large range of modelled mass (ice) flux through NEGIS, and therefore large uncertainties. This is relevant for future predications, as we don't have observations.
* * *
**Comment**

Line#: 47

This paragraph should include the statement that you make in line 221-224, making very clear to the reader you do not think a mantle plume is presently beneath NEGIS. You are simply using a plume model to generate feasible scenarios that can be tested with the model.Without the sentence at 221, It would be easy to walk away from this paper thinking you believe there is a mantle plume presently under NEGIS (which would require substantially more evidence to justify).

**Response**

Thank you this is a good point and we agree. We have changed this paragraph accordingly by including this statement earlier in the paper, LXX, and removed it from the discussion.
* * *
**Comment**

Line#: 55

How was the model changed from Schlegel to the in review paper? If you are including those modifications here, it is important that the reader know what they are, but they cannot be determined as the paper referenced is not published. This is a case where an in review citation may be acceptable, but you need to include the salient details from the paper in the text here.

**Response**

Thank you for pointing this out to us. The most important change is that the thermomechanical ice flow model is coupled to a subglacial hydrology model. We changed the sentence to include this, and keep the reference to the Smith-Johnsen et al. paper, as it is now accepted.
* * *
**Comment**

Line#: 58

Could you provide justification for your choice in sliding law here?

**Response**

This is the most commonly sliding law used in ISSM. It was used by Schlegel et al. 2015 and Smith-Johnsen et al. accepted, so to avoid a complete new model set-up with following spin-up, we decided to keep it. Instead of justifying the choice of sliding law here, we included a discussion on the implications of using this in the new caveat section of the discussion (LXX).
* * *
**Comment**

Line#: 87-88

This statement does not agree with the seismic results collected at the onset of NEGIS, where there was no apparent relationship between topography and till strength. You should reference whether or not this argument is observationally substantiated. It would be helpful to

include discussion here from Christianson et al: Christianson, K., Peters, L. E., Alley, R. B., Anandakrishnan, S., Jacobel, R. W., Riverman, K. L., ... Keisling, B. A. (2014). Dilatant till facilitates ice-stream flow in northeast Greenland. Earth and Planetary Science Letters, 401, 57–69. https://doi.org/10.1016/j.epsl.2014.05.060

**Response**

Thank you for this reference. We do compare our friction coefficient distribution to roughness observations in line X. The friction coefficient includes everything unknown at the bed, in addition to till strength. We are aware of the limitations by using this very crude and simple approach for the friction coefficient, and have included the this in the new caveat section in the discussion.
* * *
**Comment**

Line#: 101

The plumes discussed here are not very consistent with MacGregor et al 2016, who find large areas of basal melt ($> 100km$ x 100km) well upstream of NEGIS. I think the agreement between Fahnestock and MacGregor throughout the manuscript is generally overstated.

**Response**

We tried to explain that the plumes here compares well to the northeastern branch of the anomaly of MacGregor et al 2016. We have made this clearer by removing the references to figures within this paper, and instead included a plot of the MacGregor gridded dataset for our model domain in a new figure (X). Hopefully this will improve the understanding of the reader, and allow for a more direct comparison of the basal melt rates from our 970plume to the dataset.

Overall in the paper we have modified the comparison of the GHF and basal melt to previous studies, by providing more details for each comparisons.
* * *
**Comment**

Line#: 137

Clarify what you mean here, Fahnestock and MacGregor did not have identical results.

**Response**

Thank you, we mean the maximum magnitude of geothermal heat flux ($970\ mW/m^2$) proposed by Fahnestock et al. We have clarified this, and we removed the MacGregor reference.
* * *
**Comment**

Line#: 163-164

Here is an example of potentially misleading language –you show the elevated heat required by your model to initiate NEGIS. Much less heat may be required if the bed were uniformly weaker, if you included fabric evolution or imposed viscosity transitions, if the water transmissivity at the bed were lower, etc. All of the values you provide are contingent on the physical processes included in the model, the assumptions about the flow law form and parameters, and the experimental design.

**Response**

Thank you, we toned down the statement by writing "indicate" in stead of "show", and we included "in our model". In addition, we have included a caveat section in the discussion where we provide several

reasons for why we may overestimate the geothermal heat of the plume, due to model uncertainties and assumptions (friction law, shear margin softening, subglacial hydrology parameters).

However, if the bed were uniformly weaker the entire domain would speed up, resulting in an even lower surface, and increase the underestimation of ice thickness. In addition, the outlet glaciers are too fast in our model, and a uniformly weaker bed would intensify the problem. We agree that this is a very crude estimate of basal friction, and we also included this in the caveat section in the discussion.
* * *
**Comment**

Line#: 168

"metshould be "melt"

**Response**

Thanks, done
* * *
**Comment**

Line#: 173-174

This shows that plumes with a restricted extent, 50km x 50km, produce model results more consistent with the observed flow behavior in the upstream reaches of NEGIS.-something that clarifies that this is not a necessary condition for NEGIS.

**Response**

Thanks, done
* * *
**Comment**

Line#: 197-198

Perhaps change this sentence to read the geothermal heat flux needed to induce the observed upstream velocity of NEGIS in our model is 970, consistent with values presented in Fahnestock et al. (2001)."What you are stating here (and in your next sentence) is essentially "high melt water production rates are required to drive fast flow in the upstream regions of NEGIS, assuming the absence of other variations in bed strength driven by substrate heterogeneity". I think that last caveat is important to make here and elsewhere in the paper; you are forcing all of the variation to be driven by hydrology, but it need not be the only property that varies in space.

**Response**

We agree, and changed the sentence to what you suggested. However, your next statement is not exactly true, as our friction coefficient is spatially varying not uniform, and represents everything that varies at the base. We show the importance of spatially varying bed properties, by running two simulations where we have a spatially uniform friction coefficient (simulation "Uni Ctrl" and "Uni 970") where the velocity pattern is less confined and less similar to observations.

Thank you, we agree that it is important to state, that we keep everything constant in our model and only vary GHF, and try to explain the observed velocities by hydrology only despite many model assumptions and uncertainties. We stated this at the beginning of the new caveat section in the discussion, LXX.

**Comment**

Line#: 211-212

The comparison with Jarosch and Gudmundsson (2007) here seems odd, as they apply their geothermal flux anomaly over 500m. No one would argue that their anomaly could exist at the scale of your plume. However, their results do highlight something that I think you should present to your reader –substantial melt anomalies manifest in the ice sheet surface. I imagine the ice sheet surface in your models has a similar (albeit smaller)melt featureas the one in Jarosch and Gudmundsson. If so, somewhere in this work you should state that localized, substantial melt under NEGIS would be visible at the ice sheet surface, but is not apparent in altimetry data. Any discrepancy (or, if present, agreement) in the effect of basal melt on the ice surface profile must be discussed.

**Response**

Thank you, we agree and have removed this reference. In addition, Iceland is generally not a representative comparison as it is situated on a mantle plume and a spreading ridge, thus an extreme geothermal heat flux example. This is a very interesting topic. As mentioned earlier, by comparing the plume970 simulation to the Ctrl, to disentangle the direct impact of the plume, we do not see a significant local surface depression. We think this is due to the hydrology might disperse the signal, and most importantly the advection of ice redistributes and dampens the surface signal. However, we think the case would be different if the ice above the local plume was not sliding. It would be interesting to test our plume in an areas where a local mantle plume would not trigger fast flow, only local melt, to see if the surface expression would look different.

**Comment**

Line#: 218-219

This seems to imply that your results differ because you are fitting to velocities instead of temperatures, but that is not the primary factor. Greve has no constraints near the onset of NEGIS, while your study does. If the anomaly you argue for existed, Greve would have no way of knowing with the data he has available. Greve's data set is actually a much more direct measure of geothermal flux–if he had broader observational coverage it would be hard to argue with his results.

**Response**

This is true and a good point, and we removed this reasoning from the paper. We compare our GHF values to the highest estimate of Greve, and clearly state that this is from NGRIP. We have restructured this entire section and the following section to be more reader friendly.

**Comment**

Line#: 221-223

As stated earlier, this sentence should come much sooner in the paper. Without additional data, we have no means of explaining why there might be a heat flux anomaly at NEGIS, and it is not likely a modern plume.

**Response**

Thank you, as indicated earlier, we moved this sentence to the introduction LXX.

**Comment**

Line#: 227-228

MacGregor et al. have abnormally high melt rates in several places in Greenland, including over a broad region upstream of NEGIS and in SW Greenland. This citation here seems inconsistent with the statement made.

**Response**

Thank you, we removed the statement as it did not contribute to the section. We did, as mentioned, include a plot of the melt anomaly by MacGregor (Fig 6) for an improved, direct comparison of both magnitude and spatial extent.

**Comment**

Line#: 273-277

A broader discussion of the role of the friction law would be useful. What if you used a non-linear sliding law? What direction would that change your results? It would be useful for the reader to understand how the plume characteristics you describe would need to vary to reproduce NEGIS using arange of different model set-ups.

**Response**

Yes this is a caveat of our model set-up, and after your recommendation we extended the discussion of the linear friction law in the new caveat section of the discussion (LXX). We agree that the plume would change given a different model setup, and this is discussed in more detail in section starting at LXX.

**Comment**

Line#: 290

"confirms previous studies"is too strong. "is consistent with"would be better

**Response**

Thank you, we changed this to your suggestion.

**RC2: Signe Hillerup Larsen**

**Comment**

1. Structure of method and result section: a) The storyline in the experiment and result section does not match. In the results section the focus is on the study testing the hypothesis of the existence of a geothermal heat flux anomaly of 970 mW/m2. The rest of the experiments are described as sensitivity studies to this main hypothesis. This is not the story line in the experiment section.

**Response**

Thank you for noticing this, we have changed the storyline in the experiment section to match the one in the results. More specifically we removed the range of GHF in the sensitivity studies in the beginning of the experiment section. The storyline in the experiment section is now the following, first we present the 970 plume experiment, explain why we need a Ctrl, and finally we present the sensitivity simulations and explain the purpose of them. In the results we start with the Ctrl in order to explain the background values for all the simulations.
* * *
**Comment**

2. Results section: a) Presentation of results: I think it's a good idea to use the 50 m/yr contour to compare results. Maybe add some meta text in the beginning explaining that this is your approach and if possible add the observed contour line on all result plots for comparison?

**Response**

Good suggestion, we added a description of how we evaluate the performance of each model simulation using these contours in LXX. We agree, and originally tried to include both modeled and observed velocity contour in the results plot. However it was messy and too much information in one plot. We therefore decided with showing observed on the results apart from the velocity where we plotted modeled velocity contour.
* * *
**Comment**

b) In the first paragraph of the results section the Ctrl simulation is described as a way to obtain the basal melt rate, and then in the same paragraph the resulting velocity field is explained. I find this a bit confusing. Maybe just stick to the explanation about the velocity field, because the method to obtain N is already described in the methods section.

**Response**

Thank you expressing this, we removed the methods part and hope it is less confusing now. In fact, we removed all the part of this section concerning methods to avoid unnecessarily repetition.
* * *
**Comment**

3. Discussion section:
a) the discussion is purely focussed on the ice/bed interface, but I am wondering about how the resulting flow pattern depends on uncertainties within the ice such as viscosity and the fact that shear margins are not resolved by the 15km grid. Thus a short discussion of ice viscosity, shear margins and model resolution should in my opinion be included.

**Response**

Thank you, this is very good point that we did not include originally. We added a caveat section where we discuss how we could obtain similar high velocity as 970 experiment, by changing other parameters in the model and then getting away with lower geothermal heat flux values. In LXX we discuss the softening of shear margins and how we may overestimate the lateral drag. Thank you for this suggestion.
* * *
**Comment**

b) The aim is to have a model that is independent of present day observations. This is not strictly met in the way N is obtained, which is clearly explained. However, the bedmap is also based on modelling using present day velocity observations, which could bias the results, this makes the basal friction coefficient relate to observed velocities in a more diffuse way. This should also be mentioned somewhere.

**Response**

Thank you, we agree. We included this caveat in LXX .
* * *
**Comment**

4. Conclusions: a) Conclusions appear a bit too conclusive, and the authors should make an effort to make it clearer that they are aware that this is a relatively simple test of the hypothesis that a geothermal heat flux anomaly could explain the onset of NEGIS.

**Response**

We modified the conclusion and added a sentence on model caveats, allowing the reader to understand how the number presented is dependent on model uncertainties (LXX). As explained above we added a section in the discussion where we suggest other ways we could trigger fast flow of NEGIS in our model, apart from the geothermal heat flux.
* * *
**Comment**

Line#: 60-65

Effective pressure is defined in words twice.

**Response**

Thank you, we fixed that.
* * *
**Comment**

Line#: 153-154

The last sentence of the paragraph makes it sound a bit like that the 970 $mW/m^2$ experiment represents reality. Maybe just explain how the ice stream signature becomes weaker with lower forcing.

**Response**

Thanks, we toned down and included 'given our model set-up' in this statement.
* * *
**Comment**

    Line#: 199

    I am wondering if the width of the modelled ice stream could be related to model underestimating viscosity?

**Response**

This is very good point, and may explain the more diffuse modelled velocity pattern and lack of sharp gradients in the shear margins. We added your suggestion about width in the shear margin viscosity discussion, LXX, thanks.
* * *
**Comment**

    Line#: 212-213

    The sentence starting with: 970 $mW/m^2$ is only...should be moved to methods section.

**Response**

We agree that it is too late to include here. We find it more a result than a method, as this is computed by the plume model and not prescribed. We removed this statement from the discussion, as it is not important. We generally restructured the section in the discussion where we compare our findings to previous studies, and try to better explain why our values are so high.
* * *
**Comment**

    Line#: 222

    Maybe refer to Martos et al, 2018 or other paper that describes the continental passage over the Icelandic hotspot. This information should probably be included inthe introduction or methods section.

**Response**

Thank you, we agree and we moved this statement to the introduction (LXX). And for the high background geothermal heat flux due to Iceland plume we refer to Rogozhina et al. 2016 and Martos et al. 2018 (LXX) and it is also included in introduction (LXX).
* * *
**Comment**

    Line#: 281 By inverting for basal friction you not only create a basal friction map that cannot evolve in time, you also place all uncertainty from the model viscosity for example in the basal friction map.

**Response**

Yes this is true, everything uncertain in the model is blamed on the spatially varying 'bed properties'.
* * *
**Comment**

    Figure 1:

    Include the place names used in the text e.g. Storstrømmen and Zachariæ.

**Response**

Great suggestion, we included this in Figure 1c, where we introduce EGRIP and the model domain.

**Comment**

Figure 2, 3, 4 and 5:

Maybe show the observed (white) 50 m/yr contour in all the velocity plots where only the modelled contour is shown.

**Response**

As stated above, we originally tried this, but it looked so we avoided this.

**Comment**

References:

The reference to the Fox Maule paper or data is incomplete.

**Response**

Well spotted, we included journal volume and pages in this reference, thank you.

[revised manuscript text omitted]

---

## Author Response (AR2)

**Response to the editor comments**

January 3, 2020

**General response**

Dear Editor. Thank you for those additional comments. All the comments that were made have been addressed in this new version of the manuscript. See below for a summary of the modifications which are highlighted in the marked-up version of the manuscript.
On behalf of the authors,
Basile de Fleurian

**Editor Nanna Bjørnholt Karlsson**

**Comments**

Line 98: Reviewer N. Holscuh notes that the relationship between bed roughness and till strength is not straightforward. Please include a reference to the paper he suggests (Christianson et al., 2014) and add a sentence or two regarding how observations are contradictory for NEGIS.

The reference have been along with a sentence clarifying this point.

Lines 177-180: From figure 4 it is not immediately clear to me that the 775 experiment has a less good match with observed velocities compared to the 970 experiment. I suggest adding 4o to Figure 5 so the reader can more easily make a direct comparison.

The Figure and caption have been modified accordingly. The better match of simulation 970 is based on the width of the trunk of the ice stream close to its initiation where it matched better the observations.

Lines 206-207: Please modify this sentence to reflect that suggested GHF values do not agree between MacGregor et al. 2016 and Fahnestock et al., 2001 (in line with comments from N. Holscuh)

Now line 208. A sentence have been added to make this point clear.

Lines 211-214: The findings are overstated here - please rephrase. I assume what is meant is that a large area outside of the icestream with high GHF is not needed to initiate NEGIS but the sentence (inadvertently perhaps) implies that the reconstruction from MacGregor et al. 2016 is incorrect.

Now line 215. This has been rephrased and is hopefully not misleading anymore.

Lines 222-223: I disagree that basal melt and extent agree with MacGregor et al. 2016. This might be the case within the 50m/yr contour but it is not the case outside of the ice stream. Please modify this.

Now line 226. I also agree with this remark and the text have been modified to reflect that.

Lines 281-285: Would a larger, wider mantle plume combined with shear margin softening produce a good fit with observations? I suspect so. Add a sentence or two stating whether or not that might be the case.

Now 288. The softening of the shear margin and/or the use of a friction law with a higher impact of subglacial water pressure should both help to constrain the flow of the ice-stream in a more focused region. Those points should be investigated further and a few lines have been added on this point into the text.

**Minor comments**

Line 50: repetition of plume

Fixed

Line 58: Erroneous parenthesis. Should be Smith-Johnsen et al. (2019)

Fixed

Lines 143-155: It is confusing that figure 2e is discussed before figures 2b-d. Please consider reordering the subfigures in a more logical order. For example, start with the highest GHF and then add the figures for decreasing values.

The figure ordering have been modified as suggested and the references in the text are modified to reflect that.

Figure 1: Please add an outline showing where the floating parts of the domains are.

A yellow outline have been added at the grounding line and the caption is updated accordingly.

Figure 4: "k" is missing

Fixed

Figure 5: Please add more contour lines to ease comparison. For example, a white outline at 200m/yr and/or a dashed outline at 500m/yr (or whatever you find appropriate)

We tried a number of possibilities to include more comparison points. However all our attempts resulted in more cluttered figures and did not improve readability which is why we decided to stick to a single isoline for comparison.

In the manuscript, you alternate between writing "geothermal heat flux" and GHF. Please check for consistency.

Consistency has been restored by using GHF throughout the text.

[revised manuscript text omitted]